# Architecture of TAF11/TAF13/TBP complex suggests novel regulation properties of general transcription factor TFIID

Kapil Gupta[1,2], Aleksandra A Watson[3], Tiago Baptista[4,5,6,7], Elisabeth Scheer[4,5,6,7], Anna L Chambers[1], Christine Koehler[8], Juan Zou[9,10], Ima Obong-Ebong[11], Eaazhisai Kandiah[2,12], Arturo Temblador[2], Adam Round[2], Eric Forest[12], Petr Man[13,14], Christoph Bieniossek[2], Ernest D Laue[3], Edward A Lemke[8], Juri Rappsilber[9,10], Carol V Robinson[11], Didier Devys[4,5,6,7], Làszlò Tora[4,5,6,7]*, Imre Berger[1]*

[1]BrisSynBio Centre, The School of Biochemistry, Faculty of Biomedical Sciences, University of Bristol, Bristol, United Kingdom; [2]European Molecular Biology Laboratory, Grenoble, France; [3]Department of Biochemistry, University of Cambridge, Cambridge, United Kingdom; [4]Institut de Génétique et de Biologie Moléculaire et Cellulaire IGBMC, Illkirch, France; [5]Centre National de la Recherche Scientifique, Illkirch, France; [6]Institut National de la Santé et de la Recherche Médicale, Illkirch, France; [7]Université de Strasbourg, Illkirch, France; [8]European Molecular Biology Laboratory, Heidelberg, Germany; [9]Wellcome Trust Centre for Cell Biology, University of Edinburgh, Edinburgh, United Kingdom; [10]Chair of Bioanalytics, Institute of Biotechnology, Technische Universität Berlin, Berlin, Germany; [11]Physical and Theoretical Chemistry Laboratory, Oxford, United Kingdom; [12]Institut de Biologie Structurale IBS, Grenoble, France; [13]Institute of Microbiology, The Czech Academy of Sciences, Vestec, Czech Republic; [14]BioCeV - Faculty of Science, Charles University, Prague, Czech Republic

*For correspondence:
laszlo@igbmc.fr (LT);
imre.berger@bristol.ac.uk (IB)

**Abstract** General transcription factor TFIID is a key component of RNA polymerase II transcription initiation. Human TFIID is a megadalton-sized complex comprising TATA-binding protein (TBP) and 13 TBP-associated factors (TAFs). TBP binds to core promoter DNA, recognizing the TATA-box. We identified a ternary complex formed by TBP and the histone fold (HF) domain-containing TFIID subunits TAF11 and TAF13. We demonstrate that TAF11/TAF13 competes for TBP binding with TATA-box DNA, and also with the N-terminal domain of TAF1 previously implicated in TATA-box mimicry. In an integrative approach combining crystal coordinates, biochemical analyses and data from cross-linking mass-spectrometry (CLMS), we determine the architecture of the TAF11/TAF13/TBP complex, revealing TAF11/TAF13 interaction with the DNA binding surface of TBP. We identify a highly conserved C-terminal TBP-interaction domain (CTID) in TAF13, which is essential for supporting cell growth. Our results thus have implications for cellular TFIID assembly and suggest a novel regulatory state for TFIID function.

DOI: https://doi.org/10.7554/eLife.30395.001

## Introduction

Eukaryotic gene expression is a highly regulated process which is controlled by a plethora of proteins, arranged in multiprotein complexes including the general transcription factors (GTFs), Mediator and RNA polymerase II (Pol II) (*Gupta et al., 2016*; *Thomas and Chiang, 2006*). Regulated class II gene transcription is initiated by sequential nucleation of GTFs and Mediator on core promoter DNA (*Buratowski and Sharp, 1990*). The GTF TFIID is a cornerstone in this process and links cellular signaling events with regulatory DNA elements and the components of the transcription machinery (*Albright and Tjian, 2000*). A basal transcription system which supports initiation can be reconstituted with TBP and GTFs TFIIA, TFIIB, TFIIE, TFIIF and TFIIH in vitro (with TFIIA not essential if TBP repressors are not present in the system); however, TFIID is required to respond to activators (*Hampsey and Reinberg, 1999*). In mammalian cells, the promoters of virtually all protein-encoding genes are occupied by TFIID, and loss of TFIID components causes embryonic lethality (*Gegonne et al., 2012*; *Kim et al., 2005*; *Mohan et al., 2003*). TFIID subunits are thought to mediate cross-talk with epigenetic modifications on nucleosomes and regulatory DNA elements in promoter regions (*Vermeulen et al., 2007*; *Verrijzer et al., 1995*). X-ray crystallography revealed many details of TFIID components at near atomic resolution (*Gupta et al., 2016*). Cryo-electron microscopy (cryo-EM) provided essential insight into TFIID architecture and promoter DNA interaction (*Bieniossek et al., 2013*; *Cianfrocco et al., 2013*; *Louder et al., 2016*). The recent identification of a discrete TAF-containing complex in the cytoplasm of cells provided first insight into holo-TFIID assembly from preformed sub-modules, regulated by nuclear import mechanisms (*Trowitzsch et al., 2015*).

Canonical human TFIID comprises TATA-binding protein (TBP) and 13 TBP-associated factors (TAFs) (*Matangkasombut et al., 2004*; *Müller and Tora, 2014*). Furthermore, non-canonical TFIID and TAF-containing complexes have been identified regulating spermatogenesis and stem cell development (*Goodrich and Tjian, 2010*; *Maston et al., 2012*; *Müller et al., 2010*). A nuclear core-TFIID complex was identified, made up of two copies each of TAF4, 5, 6, 9 and 12 (*Bieniossek et al., 2013*; *Wright et al., 2006*). Biochemical and structural studies established the histone-fold domain (HFD) as a key TAF-TAF interaction motif within TFIID (*Gangloff et al., 2001a*). TAF3, 4, 6, 8, 9, 10, 11, 12 and 13 contain HFDs and assemble specifically into heterodimers (TAF3-10, TAF4-12, TAF6-9, TAF8-10 and TAF11-13) (*Birck et al., 1998*; *Gangloff et al., 2001b*; *Werten et al., 2002*; *Xie et al., 1996*).

TFIID recognizes core promoter DNA via its TATA-box-binding protein subunit, TBP. TBP is central to transcription regulation in eukaryotes and is the only subunit present in the transcription initiation complexes of each of the three RNA polymerases (*Koster et al., 2015*; *Thomas and Chiang, 2006*; *Tora and Timmers, 2010*). TBP consists of a highly variable N-terminal domain with less well-understood function and a conserved DNA-binding C-terminal core domain comprising two symmetrical pseudo-repeats (*Thomas and Chiang, 2006*; *Tora and Timmers, 2010*). Crystal structures of the conserved TBP core revealed a saddle-like shape with a concave DNA binding surface recognizing the minor groove of TATA-box containing DNA (*Kim et al., 1993*; *Nikolov et al., 1996*; *Nikolov et al., 1992*).

The DNA-binding activity of TBP/TFIID is tightly regulated by gene-specific co-factors that can activate or inhibit transcription (*Koster et al., 2015*; *Tora and Timmers, 2010*). The mechanism of a number of these regulatory factors has been described in molecular detail. The TFIID component TAF1 was found to associate with the concave DNA-binding surface of TBP via its N-terminal domain (TAF1-TAND), exhibiting TATA-box mimicry (*Anandapadamanaban et al., 2013*). TAF1-TAND, unstructured in isolation, was found to adopt a three-dimensional structure closely resembling the TATA-element is shape and charge distribution when bound to TBP (*Burley and Roeder, 1998*; *Liu et al., 1998*). This interaction is conserved in yeast, *Drosophila* and human (*Anandapadamanaban et al., 2013*; *Burley and Roeder, 1998*; *Liu et al., 1998*; *Mal et al., 2004*). The recent high-resolution structure of TBP bound to yeast TAF1-TAND revealed anchoring patterns in transcriptional regulation shared by TBP interactors, providing insight into the competitive multi-protein TBP interplay critical to transcriptional regulation (*Anandapadamanaban et al., 2013*). Mot1 is an ATP-dependent inhibitor of TBP/TATA-DNA complex formation (*Auble and Hahn, 1993*). Mot1 regulates the genomic distribution of TBP and was shown to influence transcription levels both positively and negatively (*Pereira et al., 2003*). Recent structural analysis revealed the molecular

mechanism of Mot1 wrapping around TBP resembling a bottle opener, with a 'latch helix' blocking the concave DNA-binding surface of TBP and acting as a chaperone to prevent DNA re-binding to ensure promoter clearance (*Wollmann et al., 2011*). Mot1 and negative co-factor 2 (NC2) are thought to cooperate in gene-specific repression of TBP activity (*Hsu et al., 2008*). The GTF TFIIA, on the other hand, competes with NC2 for TBP (*Kamada et al., 2001*; *Xie et al., 2000*) and promotes TBP/DNA interactions in a ternary TFIIA/TBP/DNA complex, facilitating formation of and stabilizing the preinitiation complex (PIC). Interaction of TFIIA with TBP results in the exclusion of negative factors that would interfere with PIC formation, and TFIIA acts as a coactivator assisting transcriptional activators in increasing transcription levels (*Bleichenbacher et al., 2003*).

Genetic and biochemical experiments suggested that the TFIIA/TBP/DNA complex is further stabilized by the histone-fold containing TFIID subunits TAF11 and TAF13 conveying the formation of a TAF11/TAF13/TFIIA/TBP/DNA assembly (*Kraemer et al., 2001*; *Lavigne et al., 1999*; *Robinson et al., 2005*). We therefore set out to investigate this putative pentameric complex in detail. Unexpectedly, we did not observe a stabilization of TFIIA/TBP/DNA by TAF11/TAF13, but found a marked destabilization of the TFIIA/TBP/DNA complex by TAF11/TAF13, resulting in the release of free DNA and the formation of a stable ternary complex formed by TAF11/TAF13 and TBP. We analyzed the TAF11/TAF13/TBP complex biochemically and structurally utilizing a comprehensive, integrative approach. We report a novel interaction of the TAF11/TAF13 dimer binding to the concave DNA-binding groove of TBP, thus excluding TATA-box containing DNA. Using pull-down experiments with immobilized TAF1-TAND, we demonstrate competition between TAF11/TAF13 and TAF1-TAND for TBP binding. We identify a novel C-terminal TBP-binding domain (CTID) within TAF13 which is highly conserved from yeast to man. We probe key residues within this TAF13 CTID by mutagenesis in vitro and in vivo in cell growth experiments, revealing a key role of this domain for viability. We contrast the TAF11/TAF13 interaction with other TBP DNA-binding groove interactors including Mot1 and discuss the implications of our findings in the context of TFIID assembly. Based on our results, we propose a novel, functional state of TFIID in transcription regulation.

## Results

### Identification of a novel TAF11/TAF13/TBP ternary complex

We set out to analyze the structure of a putative pentameric TAF11/TAF13/TFIIA/TBP/DNA complex (*Kraemer et al., 2001*; *Lavigne et al., 1999*; *Robinson et al., 2005*), with the objective to better understand the possible roles of TAF11/TAF13 in TFIID function. First, we purified human TAF11/TAF13 complex and TBP to homogeneity (*Figure 1A*). TFIIA in human cells is made from two precursor polypeptides, TFIIAαβ and TFIIAγ, with TFIIAαβ processed in vivo into two separate polypeptides, α and β, by proteolytic cleavage mediated by the protease Taspase1 (*Høiby et al., 2007*). Recombinant human TFIIA is typically produced in *E. coli* by refolding from three separate polypeptides expressed in inclusion bodies, which correspond to the native α, β and γ chains (*Bleichenbacher et al., 2003*). To facilitate recombinant human TFIIA production, we designed a single-chain construct (TFIIA$^{s-c}$) by connecting α, β and γ by flexible linkers, based on atomic coordinates taken from the crystal structure of human TFIIA/TBP/DNA complex (*Bleichenbacher et al., 2003*). TFIIA$^{s-c}$ could be produced in high amounts in soluble form in *E. coli* and purified to homogeneity without any need for refolding steps (see Materials and methods section). We stored highly purified TFIIA$^{s-c}$ at 4$^0$C, and observed the formation of needle-shaped crystals in the storage buffer after several weeks. We improved the crystals manually and determined the 2.4 Å crystal structure of TFIIA$^{s-c}$ (*Figure 1B*, *Figure 1—figure supplement 1*, *Table 1*). The crystal structure revealed a virtually identical conformation of unliganded TFIIA$^{s-c}$ as compared to TFIIA in the TFIIA/TBP/DNA complex. Moreover, the crystal structure highlighted the importance of the connecting loops we had introduced in stabilizing the three-dimensional crystal lattice (*Figure 1—figure supplement 1*). TFIIA$^{s-c}$ was active in a band-shift assay with TBP and adenovirus major late promoter (AdMLP) TATA-DNA (*Figure 1C*) similar to purified TFIIA using the classical refolding protocol (*Bleichenbacher et al., 2003*).

Next, we attempted to reconstitute the TAF11/TAF13/TFIIA/TBP/DNA complex following published procedures (*Kraemer et al., 2001*; *Robinson et al., 2005*). Titrating TAF11/TAF13 dimer to a preformed TFIIA/TBP/DNA complex had been reported to stabilize TFIIA/TBP/DNA in band-shift

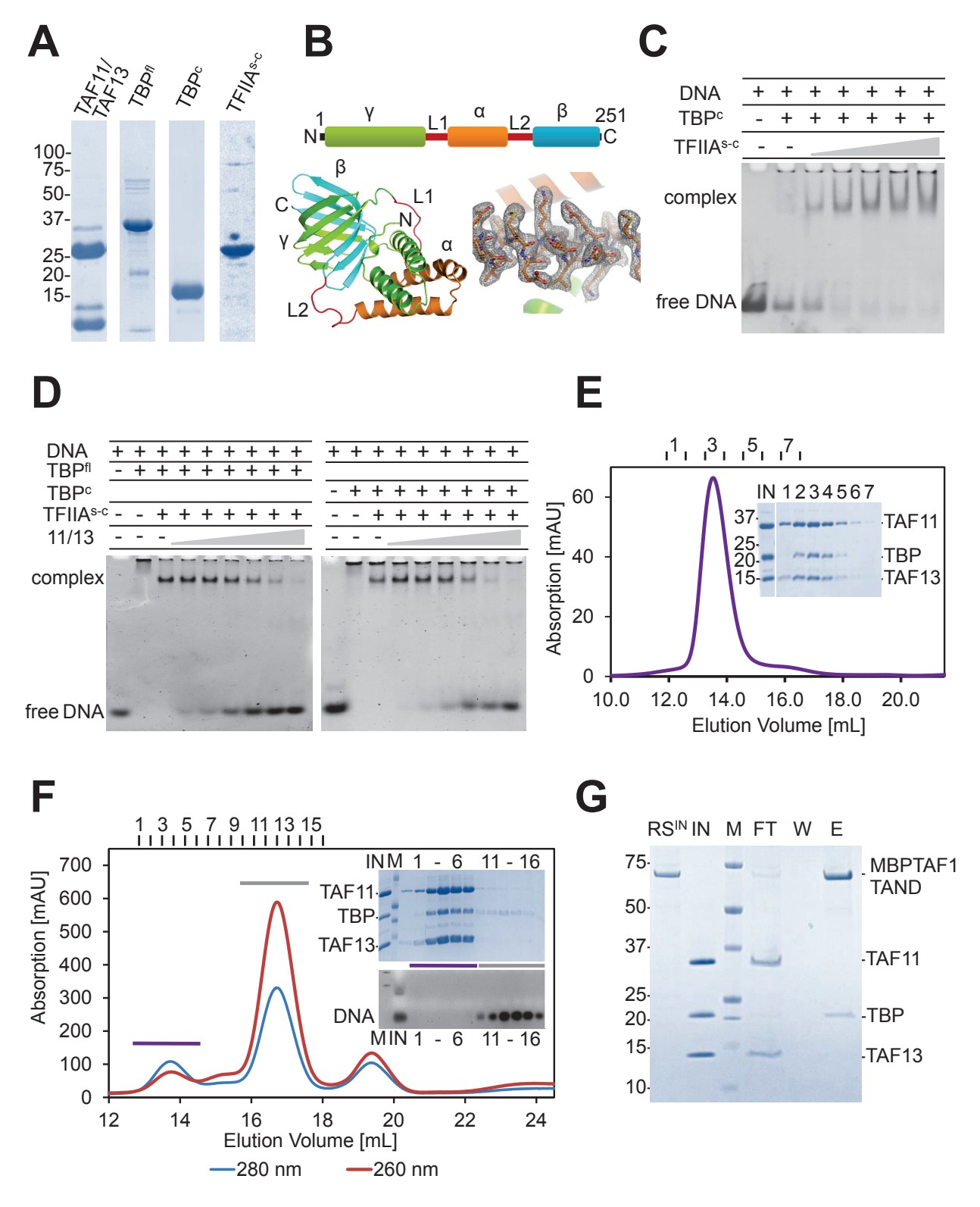

**Figure 1.** TAF11/TAF13 and TBP form ternary complex. (**A**) Human TAF11/TAF13 complex, TBP full-length (TBP^fl) and core (TBP^c), and a single-chain version of TFIIA (TFIIA^s-c) were purified to homogeneity as shown by SDS-PAGE. (**B**) TFIIA subunits α(AA2–59), β(AA302–376) and γ(AA2–110) (*Bleichenbacher et al., 2003*) were connected with linkers L1(-DGKNTANSANTNTV-) and L2(-SRAVDGELFDT-) as indicated (top). TFIIA^s-c crystallized during purification (*Figure 1—figure supplement 1*). The 2.4 Å X-ray structure is shown in a cartoon representation with a section of electron density at

*Figure 1 continued on next page*

*Figure 1 continued*

1σ(bottom). (C) TFIIA[s-c] was assayed by band-shift for activity. DNA, Adenovirus major late promoter (AdMLP) DNA. (D) Probing formation of putative pentameric TAF11/TAF13/TBPTFIIA/DNA complex by band-shift assay (*Kraemer et al., 2001*; *Robinson et al., 2005*). TAF11/TAF13 titration to the TFIIA/TBP/DNA complex results in DNA release. (E) SEC analysis reveals a stable TAF11/TAF13/TBP ternary complex. Elution fractions (1-7) were analyzed by SDS-PAGE (inset). IN, equimolar mixture of TAF11/TAF13 and TBP. No interactions were found between TAF11/TAF13/TBP complex and TFIIA (*Figure 1—figure supplement 2*). (F) TAF11/TAF13 competes with DNA for TBP binding, evidenced by SEC. Elution fractions (1-6, 11-16) were analyzed by SDS-PAGE and ethidium-bromide stained agarose gel (inset). IN, preformed TAF11/TAF13/TBP complex; DNA, AdMLP DNA; M, protein molecular weight marker and DNA ladder, respectively. (G) Immobilized human TAF1 N-terminal domain (TAF1-TAND) (*Anandapadamanaban et al., 2013*) efficiently depletes TBP from preformed TAF11/TAF13/TBP complex. RS[IN], amylose resin with MBP-tagged TAF1-TAND bound; IN, preformed TAF11/TAF13/TBP; M: protein marker; FT, flow-through fraction; W, wash fraction; E, maltose elution fraction.

DOI: https://doi.org/10.7554/eLife.30395.002

The following figure supplements are available for figure 1:

**Figure supplement 1.** Crystal structure of TFIIA[s-c] at 2.4 Å resolution.
DOI: https://doi.org/10.7554/eLife.30395.003
**Figure supplement 2.** Size exclusion chromatography (SEC) analysis of TAF11/TAF13/TBP and TFIIA[s-c].
DOI: https://doi.org/10.7554/eLife.30395.004

assays using AdMLP TATA-DNA (*Robinson et al., 2005*). Surprisingly, in our titration experiments, TAF11/TAF13 did not stabilize the preformed TFIIA/TBP/DNA complex but, in marked contrast, resulted in TAF11/TAF13-dependent release of free promoter-containing DNA in band-shift assays

**Table 1.** X-ray data collection and refinement statistics.

| | TFIIA[s-c] |
| --- | --- |
| **Data collection** | |
| Space group | P65 |
| Cell dimensions | |
| *a*, *b*, *c* (Å) | 123.3, 123.3, 34.8 |
| a, b, g (°) | 90, 90, 120 |
| Resolution (Å) | 53.4–2.4 |
| Last resolution bin (Å) | 2.52–2.38 |
| $R_{measure}$ (%) | 12.9 (64.8) |
| $I / \sigma I$ | 11.5 (2.72) |
| Completeness (%) | 99.8 (99.9) |
| Multiplicity | 6.7 (6.8) |
| **Refinement** | |
| Resolution (Å) | 40.36–2.38 (2.44–2.38) |
| No. reflections | |
| Work set | 11859 |
| Free set | 601 |
| $R_{work}$ | 0.18 (0.27) |
| $R_{free}$ | 0.24 (0.36) |
| No. of atoms | |
| Protein | 1689 |
| Water | 50 |
| r.m.s deviations | |
| Bond lengths (Å) | 0.0223 |
| Bond angles (°) | 2.088 |

*Values in parentheses are for highest resolution shell.
DOI: https://doi.org/10.7554/eLife.30395.005

(*Figure 1D*). We analyzed the possible underlying intermolecular interactions that may be formed between the individual components TAF11/TAF13, TFIIA^s-c, TBP and AdMLP TATA-DNA. We observed that TFIIA^s-c and TAF11/TAF13 did not interact in our experiments (*Figure 1—figure supplement 2*). Combining the TAF11/TAF13 dimer with TBP, in contrast, revealed a novel TAF11/TAF13/TBP complex that was stable in size exclusion experiments (*Figure 1E*). We concluded that human TAF11/TAF13 did not further stabilize the preformed TFIIA/TBP/DNA complex, but rather sequestered TBP from this complex giving rise to a novel assembly comprising TAF11, TAF13 and TBP.

## TATA-DNA and TAF1-TAND compete with TAF11/TAF13 for TBP binding

We tested competition between TAF11/TAF13/TBP formation and TBP binding to AdMLP DNA and showed that the ternary TAF11/TAF13/TBP complex remained stable in the presence of excess AdMLP DNA (*Figure 1F*). Thus, our results indicate that TAF11/TAF13 and TATA-DNA compete for at least parts of the same binding interface within TBP, and that once TAF11/TAF13 is bound to TBP, TATA-DNA binding is precluded.

TAF1 had been shown previously to bind to the DNA-binding surface of TBP via its TAND domain (*Anandapadamanaban et al., 2013*; *Burley and Roeder, 1998*; *Liu et al., 1998*). We produced human TAF1-TAND fused to maltose-binding protein (MBP) and immobilized highly purified fusion protein to an amylose column (Materials and methods). We added preformed, purified TAF11/TAF13/TBP complex in a pull-down assay using immobilized TAF1-TAND. We found that TAF1-TAND effectively sequestered TBP from the TAF11/TAF13/TBP complex, evidenced by TAF11/TAF13 eluting in the flow-through fraction. Elution by maltose, in contrast, revealed a TAF1-TAND/TBP complex (*Figure 1G*). Together, these findings substantiate our view that TAF11/TAF13, TAF1-TAND and AdMLP TATA-DNA all interact with the concave DNA-binding surface of TBP, and that the interactions are mutually exclusive.

## TAF11, TAF13, TBP form a 1:1:1 complex

Next, we set out to determine the subunit stoichiometry within the TAF11/TAF13/TBP complex, by using two complementary methods, analytical ultra-centrifugation (AUC) and native mass-spectrometry (native MS). Both methods were consistent in revealing a 1:1:1 complex (*Figure 2A,B*; *Figure 2—figure supplement 1*). Collision-induced dissociation (CID) in native MS, evidenced a TBP monomer and a TAF11/TAF13 dimer (*Figure 2B*). Of note, our control experiment using highly purified TBP evidenced a dimer in the native MS (*Figure 2—figure supplement 1*). In aggregate, our results are consistent with a ternary assembly which accommodates one copy each of TAF11, TAF13 and TBP in the complex.

## TAF11/TAF13 interacts with the concave DNA-binding surface of TBP

With the objective to elucidate the nature of the TAF11/TAF13 interaction with TBP, and to provide direct evidence that TAF11/TAF13 indeed engages the DNA-binding concave surface of TBP, we performed hydrogen-deuterium exchange/mass spectrometry (HDX-MS) (*Rajabi et al., 2015*) experiments (*Figure 2C*, *Table 2*). We analyzed unbound TAF11/TAF13 and TAF11/TAF13/TBP, and compared changes in the deuteration levels in proteolytic peptides by MS indicating the level of protection of the respective regions in TAF11, TAF13 or TBP, respectively, upon ternary complex formation. The HDX-MS results underscored that the DNA-binding surface of TBP was recognized by TAF11/TAF13, evidenced by decrease in deuteration levels which corresponds to increased protection from the solvent of peptides located in the concave surface of TBP upon TAF11/TAF13 binding. The extent of protection within TBP further indicates that the binding of TAF11/TAF13 engages both symmetric pseudo-repeats in TBP, thus spanning the entire concave interface (*Figure 2C*). Interestingly, we identified one peptide (AA residues 157–167) in TBP which evidenced an increased level of deuteration upon TAF11/TAF13/TBP complex formation in the HDX-MS experiments (*Figure 2C*). This peptide is located at the dyad relating the two pseudo-symmetric repeats in TBP. We interpret this result as an indication, that this particular region within TBP is more protected in a presumed TBP dimer which dissociates when TAF11/TAF13 is binding and the 1:1:1 complex is formed. Our HDX-MS experiments provide direct evidence that TAF11/TAF13 engage the concave

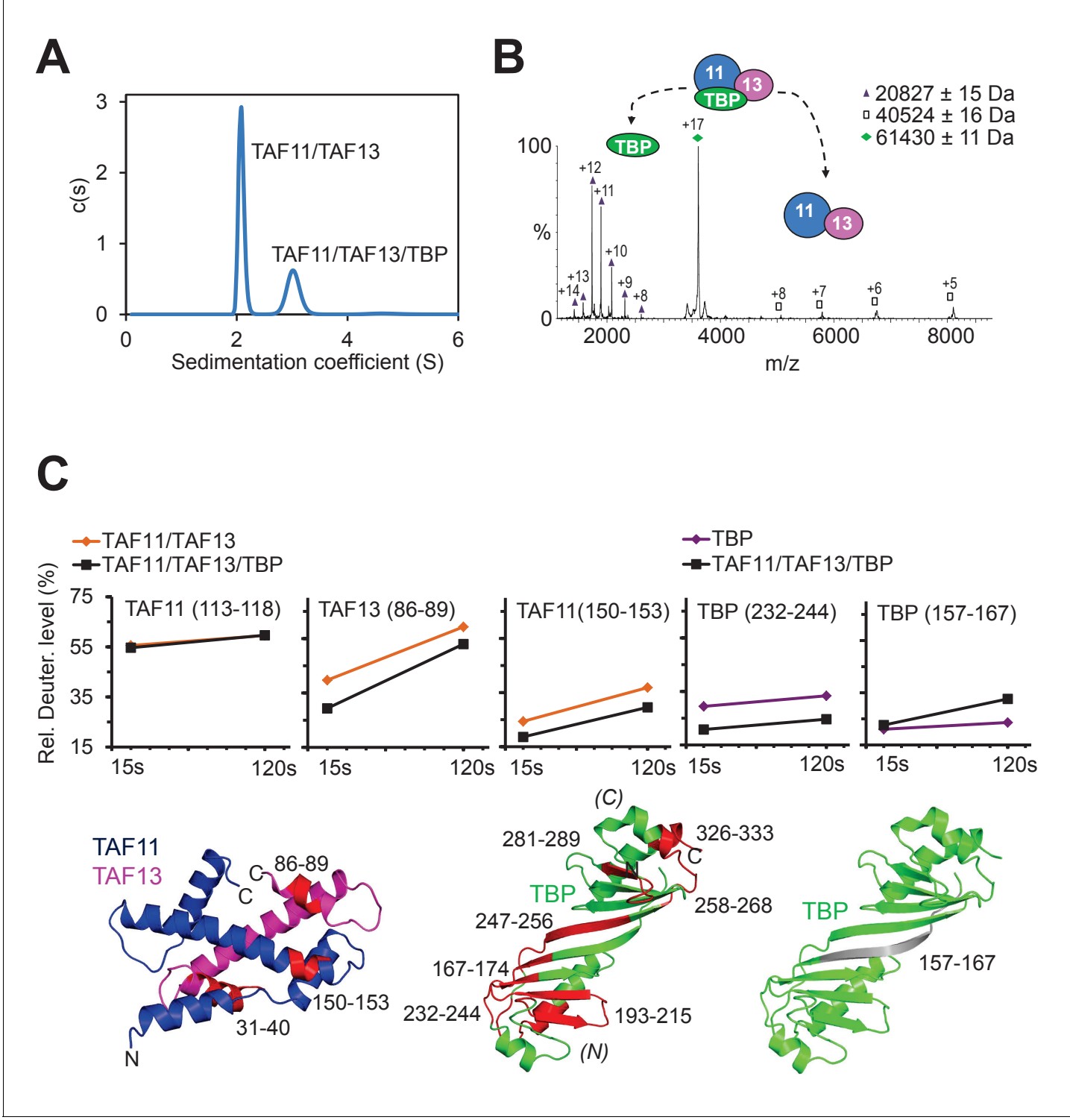

**Figure 2.** TAF11/TAF13/TBP interactions. (**A**) Sedimentation velocity analytical ultracentrifugation (AUC) experiments reveal a major peak consistent with a 1:1:1 TAF11/TAF13/TBP ternary complex. The second, smaller peak corresponds to excess TAF11/TAF13. (**B**) Native mass spectrometry of the TAF11/TAF13/TBP complex confirms 1:1:1 stoichiometry. Collision induced dissociation (CID) results in TBP monomer and TAF11/TAF13. Experimental masses are provided (inset). Calculated masses: 20659 Da (TBP); 40691 Da (TAF11/TAF13); 61351 Da (TAF11/TAF13/TBP). Mass spectra of the TAF11/TAF13/TBP complex and TBP dimer are shown in *Figure 2—figure supplement 1*. (**C**) TAF11/TAF13/TBP and TAF11/TAF13 were analyzed by hydrogen-deuterium exchange/mass spectrometry (HDX-MS) (*Rajabi et al., 2015*). Changes in the deuteration level of selected peptides in TAF11, TAF13 or TBP are depicted in diagrams (top row). Peptides protected upon ternary complex formation are coloured in red in cartoon representations

*Figure 2 continued on next page*

*Figure 2 continued*

of TAF11/TAF13 and TBP (bottom row). One peptide in TBP (grey, far right) becomes more accessible, hinting at disassembly of a TBP dimer when TAF11/TAF13 binds (*Figure 2—figure supplement 1*). All peptides implicated in TAF11/TAF13-binding map to the concave DNA-binding surface of TBP. (N) and (C) indicate N- and C-terminal TBP stirrups, respectively.

DOI: https://doi.org/10.7554/eLife.30395.006

The following figure supplement is available for figure 2:

**Figure supplement 1.** Native mass spectrometry (MS) analysis of TAF11/TAF13/TBP complex.

DOI: https://doi.org/10.7554/eLife.30395.007

**Table 2.** Peptide deuteration level changes upon complex formation in HDX-MS

**TAF11**

| Residues numbers | Sequence | $\Delta\%D$ (15 s)[†] | $\Delta\%D$ (120 s)[†] |
|---|---|---|---|
| 93–96 | EKKQ | −8.465 | −7.309 |
| 105–109 | KMQIL | −7.014 | |
| 150–153 | VVIA | | −7.295 |

TAF13

| Residues numbers | Sequence | $\Delta\%D$ (15 s)[†] | $\Delta\%D$ (120 s)[†] |
|---|---|---|---|
| 14–31 | NEEIGGGAEGGQGKRKRL | −7.507 | −7.192 |
| 32–35 | FSKE | −8.794 | |
| 36–40 | LRCMM | −7.0 | |
| 86–88 | IVF | −8.598 | |
| 86–89 | IVFL | −10.366 | |
| 97–104 | FARVKDLL | | −7.223 |
| 116–124 | AFDEANYGS | 10.89 | 8.721 |

TBP

| Residues numbers | Sequence | $\Delta\%D$ (15 s)[†] | $\Delta\%D$ (120 s)[†] |
|---|---|---|---|
| 157–167* | IVPQLQNIVST | | 9.038 |
| 167–174 | TVNLGCKL | −9.675 | −10.171 |
| 193–197 | FAAVI | −14.068 | −12.267 |
| 197–208 | IMRIREPRTTAL | −7.785 | |
| 199–208 | RIREPRTTAL | −9.661 | |
| 209–215 | IFSSGKM | −8.655 | −8.75 |
| 232–244 | KYARVVQKLGFPA | −21.337 | −18.382 |
| 233–242 | YARVVQKLGF | −7.128 | |
| 247–252 | LDFKIQ | | −7.271 |
| 250–256 | KIQNMVG | −8.913 | |
| 259–266 | DVKFPIRL | −11.932 | −10.109 |
| 259–268 | DVKFPIRLEG | −13.79 | −11.479 |
| 260–266 | VKFPIRL | −10.393 | |
| 260–268 | VKFPIRLEG | −16.066 | −13.002 |
| 281–287 | PELFPGL | | −7.811 |
| 285–289 | PGLIY | −8.061 | −7.339 |
| 326–335 | PILKGFRKTT | −7.077 | −7.902 |

*TBP peptide with increasing deuteration level upon complex formation.

[†]Peptides exhibiting changes in deuteration level ≥7% are shown.

DOI: https://doi.org/10.7554/eLife.30395.008

DNA-binding surface of TBP, in excellent agreement with our above described biochemical experiments involving TAF11/TAF13, TBP, AdMLP DNA and TAF1-TAND.

## Architecture of the TAF11/TAF13/TBP complex

We proceeded to determine the architecture of the TAF11/TAF13/TBP complex by using a comprehensive, integrative multi-parameter approach. We utilized the available crystal structure of TBP (*Nikolov et al., 1992*) as well as the crystal structure of the globular histone-fold containing domains of the TAF11/TAF13 dimer (*Birck et al., 1998*) and combined these atomic coordinates with our results from biochemical and biophysical experiments, including small angle X-ray scattering (SAXS) experiments (*Figure 3—figure supplement 1*, *Table 3*). We acquired distance constraints to define our structural model by carrying out cross-linking/mass-spectrometry (CLMS) experiments using two different approaches. We first carried out a series of CLMS experiments using the cross-linker bis(sulfosuccinimidyl)-suberate (BS3) (*Figure 3—figure supplement 2*; *Table 4*). BS3 cross-links primary amines on the side chain of lysine residues and the N-terminus of polypeptide chains. Inclusion of the BS3 CLMS derived distance constraints into our calculation already evidenced that the TAF11/TAF13 engaged the concave DNA-binding surface of TBP. In addition, we carried out site-specific UV-induced CLMS experiments utilizing MultiBacTAG (*Koehler et al., 2016*), a method we recently developed to unlock protein complex chemical space (*Figure 3—figure supplement 3*). MultiBac-TAG relies on genetic code expansion (GCE) and employs a modified MultiBac baculoviral genome into which we engineered expression cassettes encoding for the orthogonal pyrolysine tRNA

**Table 3.** Data collection and refinement statistics SAXS.

|  | TAF11/TAF13/TBP | TAF11/TAF13 |
| --- | --- | --- |
| **Data collection parameters** | | |
| Beamline | ESRF-BM29 | ESRF-BM29 |
| Beam size at sample | ~700 μm x 700 μm | ~700 μm x 700 μm |
| Wavelength (Å) | 0.931 | 0.931 |
| S range (Å−1) | 0.003–0.497 | 0.003–0.497 |
| Concentration range (mg ml-1) | 0.3–7.11 | 0.53–7.48 |
| Temperature (°C) | 4 | 4 |
| Beamline | ESRF-BM29 | ESRF-BM29 |
| Beam size at sample | ~700 μm x 700 μm | ~700 μm x 700 μm |
| Wavelength (Å) | 0.931 | 0.931 |
| Structural parameters* | | |
| I(0) (arbitrary units) [from P(r)] | 49.63 | 43.65 |
| Rg (Å) [from P(r)] | 41 | 41.2 |
| I(0) (arbitrary units) (from Guinier) | 50.21 ± 0.33 | 44.15 ± 0.08 |
| Rg (Å) (from Guinier) | 40.9 ± 0.6 | 40.3 ± 3.6 |
| Dmax (Å) | 160 | 140 |
| Porod volume estimate (Å3) | 120110 | 89850 |
| Molecular mass Mr [from porod volume] | 70.65 kDa | 52.91 kDa |
| I(0) (arbitrary units) [from P(r)] | 49.63 | 43.65 |
| Rg (Å) [from P(r)] | 41 | 41.2 |
| I(0) (arbitrary units) (from Guinier) | 50.21 ± 0.33 | 44.15 ± 0.08 |
| Rg (Å) (from Guinier) | 40.9 ± 0.6 | 40.3 ± 3.6 |
| Dmax (Å) | 160 | 140 |
| Porod volume estimate (Å3) | 120110 | 89850 |
| Molecular mass Mr [from porod volume] | 70.65 kDa | 52.91 kDa |

*Reported for experimental merged data.
DOI: https://doi.org/10.7554/eLife.30395.009

**Table 4.** Cross-links observed by BS3 CLMS.

**TAF11-TBP**

| TAF11 residue | TBP residue | No. of matches | Highest score |
|---|---|---|---|
| K97 | K293 | 5 | 9.416 |
| K85 | K293 | 2 | 8.818 |
| K82 | K239 | 2 | 8.314 |
| K135 | K191 | 1 | 8.085 |
| K82 | K293 | 2 | 7.926 |
| K82 | K191 | 1 | 7.684 |
| K131 | K293 | 1 | 6.928 |
| K83 | K232 | 1 | 6.124 |
| K95 | K293 | 1 | 4.145 |
| (9 unique links) | | | |

TAF13-TBP

| TAF13 residue | TBP residue | No. of matches | Highest score |
|---|---|---|---|
| K34 | K177 | 1 | 10.578 |
| K34 | K191 | 1 | 7.856 |
| K101 | K232 | 1 | 7.584 |
| K115 | K191 | 1 | 7.519 |
| K115 | K177 | 1 | 5.742 |
| (5 unique links) | | | |

TAF11-TAF13

| TAF11 residue | TAF13 residue | No. of matches | Highest score |
|---|---|---|---|
| K195 | K96 | 2 | 12.795 |
| K85 | K101 | 2 | 12.014 |
| K204 | K96 | 3 | 11.053 |
| K89 | K101 | 5 | 10.98 |
| K204 | K101 | 6 | 10.909 |
| K131 | K34 | 3 | 9.627 |
| K82 | K92 | 2 | 9.321 |
| K197 | K96 | 2 | 9.308 |
| K131 | K111 | 10 | 9.226 |
| K135 | K111 | 2 | 8.371 |
| K131 | K115 | 1 | 8.326 |
| K207 | K92 | 1 | 8.292 |
| K82 | K96 | 1 | 7.981 |
| K85 | K115 | 1 | 7.943 |
| K94 | K101 | 2 | 7.934 |
| K97 | K101 | 1 | 7.867 |
| K204 | K92 | 4 | 6.872 |
| K85 | K92 | 1 | 6.817 |
| K82 | K101 | 1 | 6.437 |
| K206 | K92 | 1 | 5.77 |
| K204 | S74 | 1 | 3.42 |
| K206 | S74 | 1 | 3.42 |
| (22 unique links) | | | |

DOI: https://doi.org/10.7554/eLife.30395.010

(tRNA^{Pyl})/tRNA synthetase (PylRS) pair from *Methanosarcina mazei*. Infection of insect cell cultures with a MultiBacTAG virus containing TAF11/TAF13 mutants harboring an AMBER stop codon resulted in efficient incorporation of the UV-activatable amino acid diazirin lysine (DiAzK) provided in the culture medium, leading to site-specific cross-links upon UV irradiation of the reconstituted complex (*Koehler et al., 2016*) (*Figure 3—figure supplement 3*).

Our final three-dimensional TAF11/TAF13/TBP ternary complex model accommodated more than 90% of all experimental constraints and evidenced a tight association of the TAF11/TAF13 histone-fold regions with the concave surface of TBP, giving rise to a compact structure (*Figure 3*, *Figure 3—figure supplement 4*). To validate our approach, we carried out calculations using alternative starting models. For instance, we rotated TBP by $180^0$ around its axes to artificially expose the convex surface to TAF11/TAF13, or, alternatively, to reverse the location of the N- and C-terminal stir-ups of TBP (data now shown). These alternative calculations were far inferior in accommodating experimental spatial and distance restraints, in addition to being inconsistent with our biochemical data, thus substantiating our TAF11/TAF13/TBP structural model.

## A highly conserved domain within TAF13 is required for TAF11/TAF13 interaction with TBP and for supporting cell growth

In our TAF11/TAF13/TBP complex, the HF domains of TAF11 and TAF13 invade the concave DNA-binding surface of TBP. Moreover, the structural model conveys that the C-terminal extension abutting the histone fold domain of TAF13 may play a prominent role in stabilizing the interaction with TBP. We analyzed TAF13 primary sequences from yeast to human (*Figure 4*). Sequence alignments revealed a very high degree of sequence conservation in this C-terminal TAF13 domain, with key residues virtually identical in all TAF13 proteins analyzed (*Figure 4A*). Based on our observation, we mutated these signature residues in the human TAF13 C-terminal region and analyzed the effect of the mutations on TAF11/TAF13/TBP complex formation. In particular, we analyzed two TAF13 mutants, A and B (*Figure 4*). In Mutant A, we substituted with alanine conserved residues located in the center of the TAF11/TAF13/TBP interface, while in Mutant B we changed conserved amino acid residues located more to the periphery (*Figure 4—figure supplement 1A*). Both TAF13 mutants readily formed dimers with TAF11. SEC experiments evidenced impairment of TBP interaction with both mutants, but with notable differences. In case of Mutant A, TBP interaction was completely abolished and the TAF11/TAF13/TBP complex was not detected. In case of Mutant B, on the other hand, TBP interaction was likewise diminished, however, residual TAF11/TAF13/TBP complex formation was clearly observed (*Figure 4B*). Based on this, we generated *Saccharomyces cerevisiae* (sc) Taf13 mutants containing identical mutations in their CTID as the corresponding human TAF13 Mutant A and Mutant B proteins, respectively. With plasmids expressing these scTaf13 mutants (scMutant A and scMutant B), we carried out in vivo rescue experiments in yeast, using previously described temperature-sensitive *Taf13* mutant strains (*Shen et al., 2003*; *Lemaire and Collart, 2000*). While plasmids expressing either wild-type (WT) or Mutant B Taf13 proteins supported growth at the non-permissive temperature (both on solid media and in suspension cell culture at close to wild-type levels), Mutant A did not rescue the lethal phenotype at 37°C, evidencing that the conserved mutations in this mutant effectively arrested cell growth (*Figure 4C*).

We corroborated our results by means of a novel Taf13 degron yeast strain (*Warfield et al., 2017*). In this strain, endogenous Taf13 is fused to an auxin-inducible degron (AID) tag resulting in Taf13-AID depletion upon addition of a chemical, indole-3-acetic acid (IAA). The plasmids expressing wild-type or Mutant B Taf13 supported growth after IAA addition, while expression of Mutant A lead to cell growth arrest (*Figure 4D*).

Next, we investigated whether the effects we observed could be related to compromised TFIID integrity which may have lost the mutated proteins. We performed co-immunoprecipitations (co-IPs) of hemagglutinin (HA)-tagged wild-type and mutant Taf13 proteins, and used specific antibodies to probe for the presence of Tafs and TBP in our co-IP experiments (*Figure 4E*). All TFIID subunits tested were equally present in co-IPs of HA tagged wild-type Taf13, as well as Mutant A and Mutant B. We conclude that the mutations we introduced did not cause noticeable dissociation of Taf13 from TFIID. Taken together, our observations consistently suggest that the amino acids in TAF13 CTID, which when mutated abolish TAF11/TAF13 interactions with TBP, are required for a functional TFIID complex in vivo.

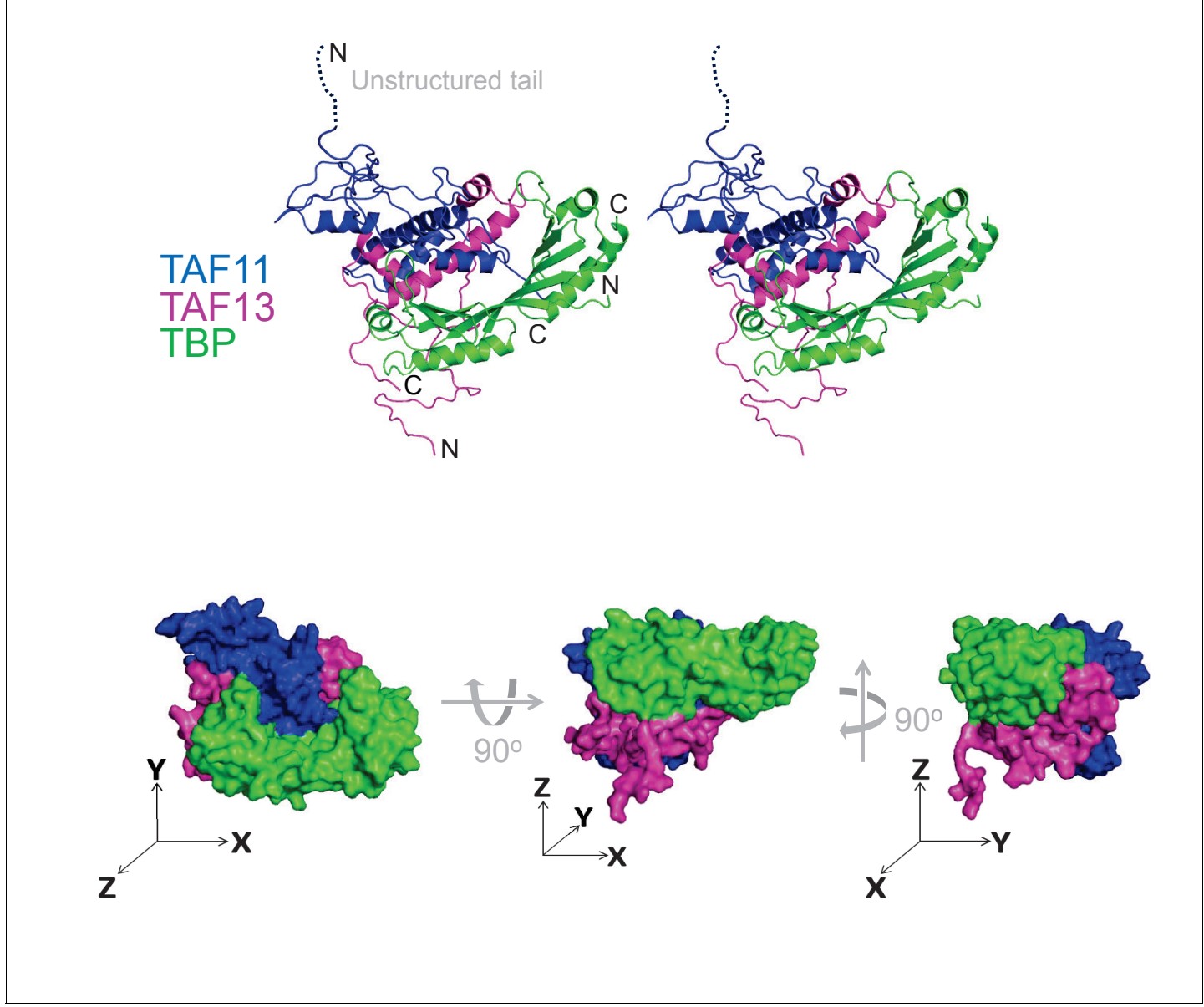

**Figure 3.** Architecture of TAF11/TAF13/TBP complex. TAF11/TAF13/TBP complex architecture was determined by using an integrative multi-parameter approach. We utilized the crystal structure of TBP (*Nikolov et al., 1996*) as well as the crystal structure of the TAF11/TAF13 dimer (*Birck et al., 1998*) combined with our native MS, SAXS, AUC and HDX-MS results as well as distance constraints from CLMS experiments (*Figure 3—figure supplements 1–3*). The structure of the TAF11/TAF13/TBP ternary complex is shown in a cartoon representation in stereo (top) and as a space filling model (devoid of unstructured regions) in three views (bottom). Three axes (x, y, z, drawn as arrows) illustrate the spatial relation between the views. TAF11 is colored in blue, TAF13 in magenta and TBP in green. This model satisfies >90% of the experimental constrains (*Figure 3—figure supplement 4*, *Tables 2* and *4*).

DOI: https://doi.org/10.7554/eLife.30395.011

The following figure supplements are available for figure 3:

**Figure supplement 1.** Small angle X-ray scattering (SAXS).

DOI: https://doi.org/10.7554/eLife.30395.012

**Figure supplement 2.** Cross-linking/mass spectrometry (CLMS).

DOI: https://doi.org/10.7554/eLife.30395.013

**Figure supplement 3.** Site-specific cross-linking of TAF11 and TAF13 by Genetic Code Expansion (GCE).

DOI: https://doi.org/10.7554/eLife.30395.014

**Figure supplement 4.** Mapping CLMS and HDX-MS data on TAF11/TAF13/TBP complex.

DOI: https://doi.org/10.7554/eLife.30395.015

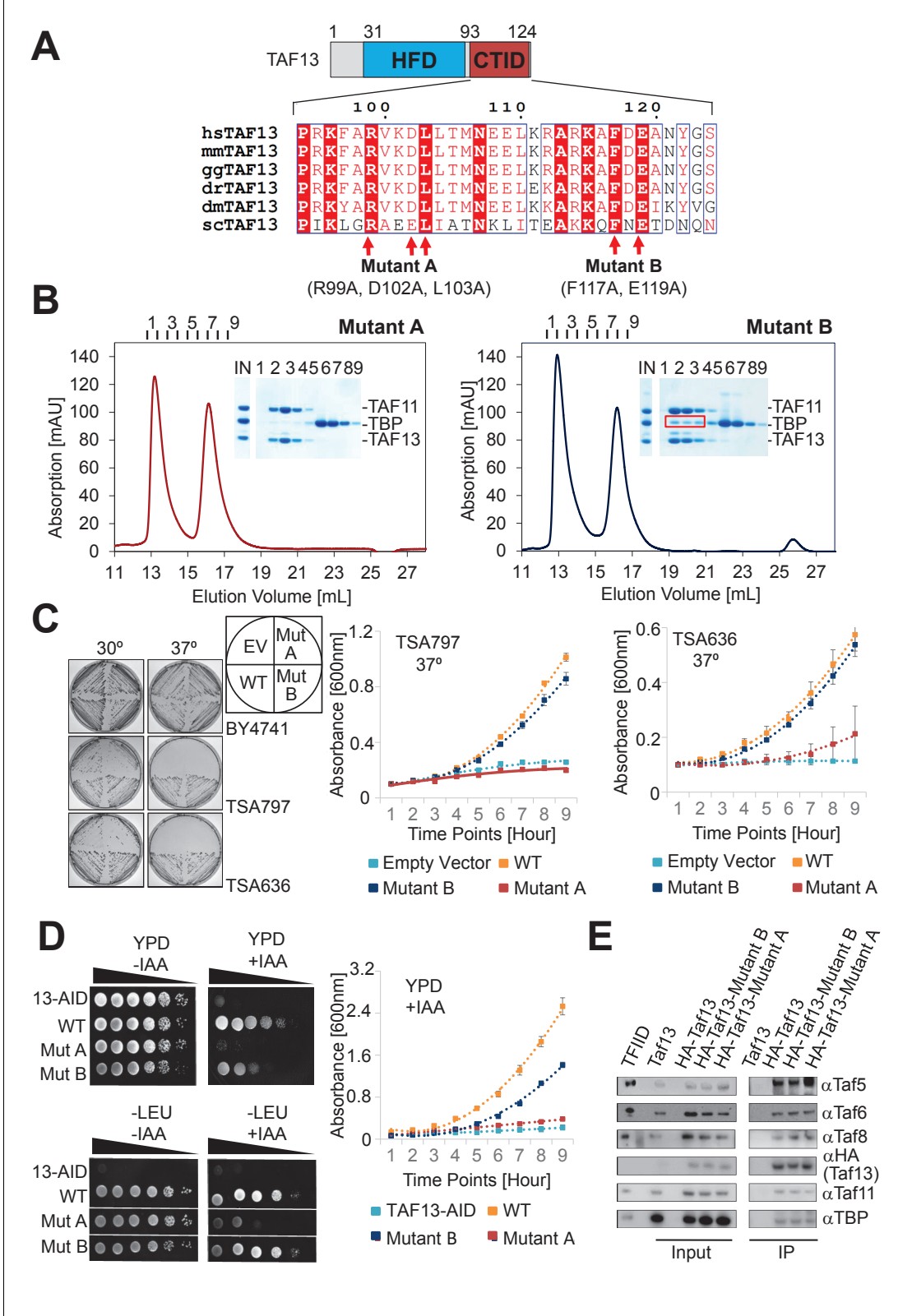

**Figure 4.** Highly conserved C-terminal TBP-interaction domain (CTID) in TAF13 required for survival. (**A**) Sequence alignments reveal a highly conserved C-terminal TBP interaction domain (CTID) in TAF13 comprising virtually identical signature residues in TAF13 from yeast to humans. Residues that were mutated in the CTID of TAF13 are indicated by arrows, giving rise to two mutant TAF13 proteins (Mutant A, **B**). The locations of the mutated residues in TAF11/TAF13/TBP are illustrated in *Figure 4—figure supplement 1A*. (**B**) SEC analysis demonstrates complete abolition of the TBP binding by TAF11/

*Figure 4 continued on next page*

*Figure 4 continued*

TAF13 in case of Mutant A. In case of Mutant B, residual interaction with TBP is observed (marked by red box). Elution fractions (1-8) were analyzed by SDS-PAGE (inset). IN, equimolar mixture of TAF11/TAF13 and TBP. (**C**) Cell growth experiments in yeast containing temperature-sensitive (ts) Taf13 on solid media plates at permissive (30°C) and non-permissive (37°C) temperatures are shown on the left. EV, empty vector; WT, wild-type Taf13; MutA, MutB, Taf13 mutants A and B; TSA797, TSA636, yeast strains harboring distinct temperature-sensitive Taf13 mutants (*Shen et al., 2003*; *Lemaire and Collart, 2000*). Corresponding absorbance plots displaying growth curves of temperature-sensitive strains in liquid media at the non-permissive temperature (37°C) are provided on the right. Polynomial fits are shown as dotted lines. Standard errors of mean (SEM) are shown as bars. The corresponding growth curves for strain BY4741 used as a control, are shown in *Figure 4—figure supplement 1B*. (**D**) Cell growth experiments in yeast containing Taf13 fused to an auxin-inducible degron tag (AID) are shown in spot assays on solid media plates (YPD, -LEU) on the left, in presence or absence of indole-3-acetic acid (IAA) which activates Taf13-AID depletion. 13-AID, Taf13 degron-tag fusion (*Warfield et al., 2017*); YPD, yeast total media; -LEU, synthetic drop-out media. Corresponding absorbance plots displaying growth curves in presence of IAA are shown on the right. Absorbance plots in absence of IAA are provided in *Figure 4—figure supplement 1C*. (**E**) Western blots from co-immunoprecipitations (co-IPs) from yeast are shown of HA-tagged wild-type and mutant Taf13 proteins, probed with specific antibodies against Taf5, Taf6, Taf8, Taf11, TBP and the HA tag on Taf13. Purified yeast holo-TFIID (marked as TFIID) and extract from yeast transformed with untagged wild-type Taf13 (marked Taf13) were used as controls. All TFIID subunits probed are equally present in all HA co-IPs.

DOI: https://doi.org/10.7554/eLife.30395.016

The following figure supplement is available for figure 4:

**Figure supplement 1.** TAF11 CTID mutant studies (**A**).
DOI: https://doi.org/10.7554/eLife.30395.017

## Co-immunoprecipitation experiments reveal cytoplasmic TAF11/TAF13 and TBP dynamics in nuclear holo-TFIID

We recently demonstrated that human TFIID assembly involves preformed cytosolic and nuclear sub-modules (*Trowitzsch et al., 2015*), and we now asked whether the human TAF11/TAF13/TBP complex would likewise represent such a sub-assembly. To this end, we performed co-immunoprecipitations (co-IPs) from HeLa cell cytosolic and nuclear extracts using an anti-TAF11 antibody (*Figure 5*). We found dimeric TAF11/TAF13 complex in the cytosol representing the complete HF pair. We could not detect TBP in cytosolic co-IPs, however, our experiments evidenced TAF7 association with cytoplasmic TAF11/TAF13. The anti-TAF11 co-IP from nuclear extract, in contrast, contained all TFIID components. Surprisingly, normalized spectral abundance factor (NSAF) (*Zybailov et al., 2007*) analyses of several distinct anti-TFIID IPs (i.e. anti-TAF1 and anti-TAF7) from NE indicate that less than half of the nuclear TFIID specimens appear to contain stably bound TBP, implying considerable dynamics in TBP association with TFIID in the nucleus, possibly regulated by TBP containing TFIID submodules such as TAF11/TAF13/TBP.

## Discussion

Given the central role of TBP in eukaryotic transcription, it is not surprising that the activity of TBP is highly regulated, both positively and negatively, by GTFs, cofactors and gene-specific activators. In addition to TFIIA, numerous protein factors were identified to regulate the DNA-binding capability of TBP. Several of these factors interact directly with the concave DNA-binding surface and are capable of displacing TATA-box containing promoter DNA. The structures of the TAF1 N-terminal domains from *Drosophila* and yeast were determined, exhibiting TATA-box mimicry when bound to TBP (*Figure 6*, *Figure 6—figure supplement 1*). TBP and BTAF1 in mammals, or TBP and Mot1p in yeast, form a heterodimeric complex called B-TFIID (*Auble and Hahn, 1993*). Mot1p was shown to utilize a flexible loop to target the DNA-binding surface of TBP thus precluding TATA-DNA binding (*Wollmann et al., 2011*). In our present study, we analyzed the interactions amongst TAF11/TAF13, TBP, TFIIA and TATA-box containing promoter DNA. TAF11 and TAF13 form a tight dimeric complex held together by pairing of the histone fold domains contained within these TAFs (*Birck et al., 1998*). Existing data conveyed a putative TAF11/TAF13/TFIIA/TBP/DNA complex which may represent a molecular building block in early stage preinitiation complex formation. In marked contrast to previous reports, our careful and well-calibrated titration of TAF11/TAF13 to preformed TFIIA/TBP/DNA did not result in a stabilization of the TFIIA/TBP/DNA complex. Rather, we observed that titration of TAF11/TAF13 resulted in the binding of TAF11/TAF13 to TBP and the release of free promoter DNA. Dissection of the underlying molecular interactions revealed a stable ternary complex

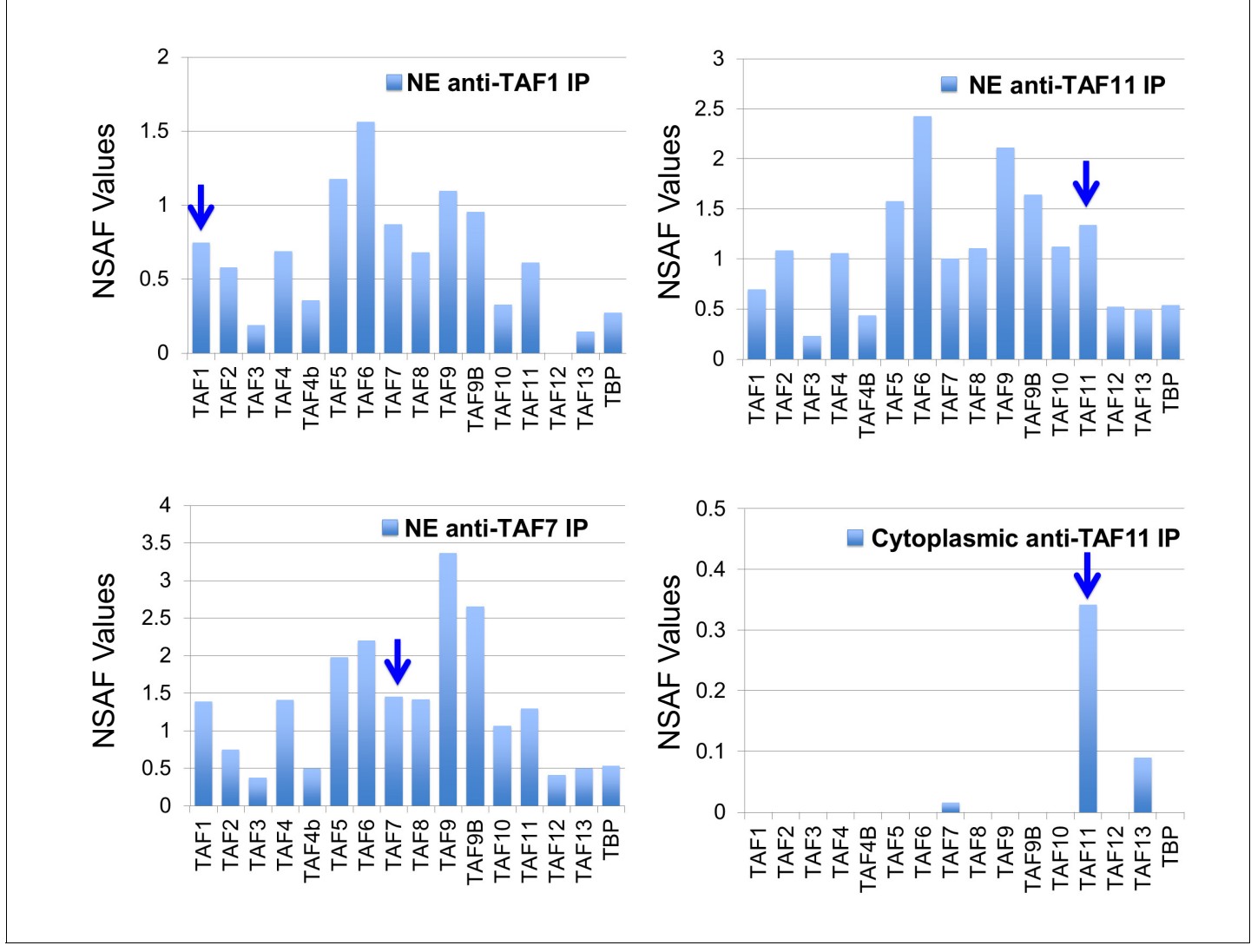

**Figure 5.** Human TAF/TFIID co-immunoprecipitations. Orbitrap mass spectroscopic analyses of proteins co-immunoprecipitated from nuclear (NE) or cytoplasmic HeLa cell extracts using mouse monoclonal antibodies against the indicated TAFs. The stoichiometry of the TAFs and TBP in the purified TFIID complexes was calculated by determining normalized spectral abundance factors (NSAFs) (*Sardiu et al., 2008*; *Zybailov et al., 2007*). Each column is the average of three independent MS runs. Blue arrows indicate the bait in each immunoprecipitation.

DOI: https://doi.org/10.7554/eLife.30395.018

comprising the TAF11/TAF13 HF pair and TBP. Our comprehensive multi-parameter approach revealed a compact 3-D structure in which TAF11/TAF13 bound tightly to the concave DNA-binding surface of TBP, fully consistent with our observations that TAF11/TAF13 could displace TATA-box containing DNA from a TBP/DNA complex. In addition, we identified in our experiments a novel C-terminal domain within TAF13 that is essential for binding to TBP, and moreover markedly conserved throughout evolution. We generated TAF13 mutants and could demonstrate that mutations of key residues within this highly conserved domain, while not perturbing either TAF13/TAF11 interactions, or holo-TFIID integrity in our experiments, had a profound effect on TBP binding in vitro and cell growth in vivo, effectively resulting in cell growth arrest. Taken together, our results indicate that the interaction of TAF11/TAF13 with TBP supports viability. Furthermore, we also demonstrated that TAF1-TAND and TAF11/TAF13 can compete for TBP binding. In summary, we provide compelling evidence that TAF11/TAF13, TATA-box DNA and TAF1-TAND share the same interaction interface in TBP. Careful inspection of the molecular modes by which different interactors engage the concave DNA-binding surface of TBP reveals that TAF11/TAF13 is unique in spanning the entire

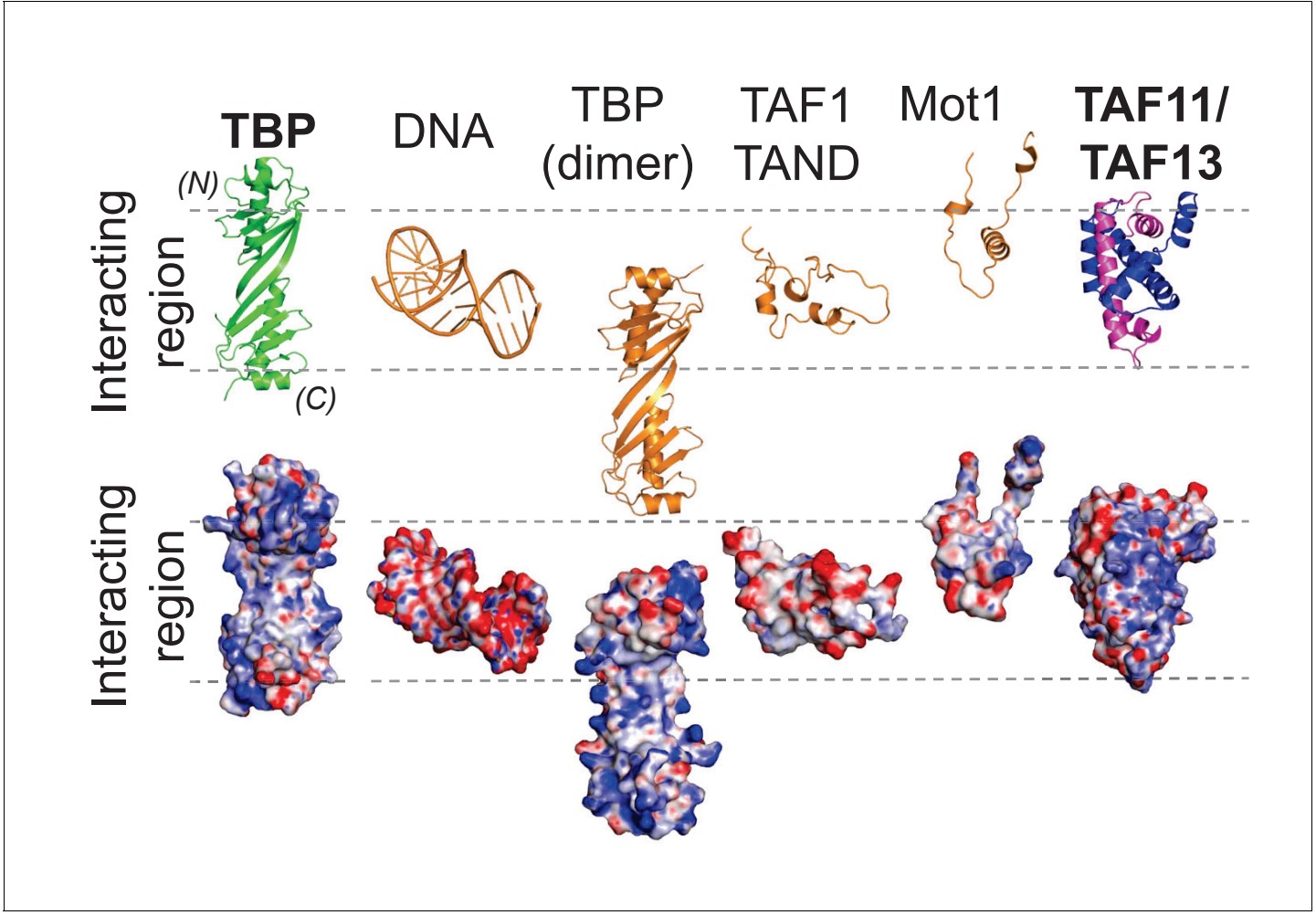

**Figure 6.** Distinct modes of TBP binding involving the concave DNA-binding surface. The interaction interfaces of TBP binders are shown in a cartoon representation (top). Interactors shown are TATA-box DNA and protein interactors including the TAF11/TAF13 dimer. The binding modes are further illustrated using space-filling models depicting the corresponding electrostatic surface potentials (bottom). The interacting region representing the concave DNA-binding surface of TBP is delimited by dashed lines. Structures shown are TBP on one hand, as well as TATA-DNA (PDB ID 1CDW), a second copy of TBP from the crystal structure of uniganded TBP (PDB ID 1TBP), TAF1-TAND (PDB ID 1TBA), Mot1 (PDB ID 3OC3) and TAF11/TAF13, respectively, on the other. (N) and (C) indicate N- and C-terminal TBP. The 'TATA-box mimicry' by TAF1-TAND in shape and charge distribution is evident. TAF11/TAF13 engage the entire concave DNA-binding surface of TBP including the stir-ups.

DOI: https://doi.org/10.7554/eLife.30395.019

The following figure supplement is available for figure 6:

**Figure supplement 1.** TAF1-TAND TATA-box mimicry in *Drosophila* and Yeast.
DOI: https://doi.org/10.7554/eLife.30395.020

concave groove including the stir-ups of the saddle-shape adopted by TBP, with a solvent-excluded surface comparable to TATA-box DNA binding to TBP (*Figure 6*, *Table 5*).

Holo-TFIID is thought to exist in distinct structural states, based on cryo-EM analyses (*Cianfrocco et al., 2013*; *Louder et al., 2016*). In the canonical state, TAF1 is proposed to associate via its TAND domain to the DNA-binding surface of TBP thereby inhibiting TBP/TFIID binding to TATA-box containing core promoter DNA. In the activated state, TFIID was proposed to undergo major conformational rearrangements, likely involving interactions with transcriptional activators, thus unmasking TBP to promote DNA binding stabilized by TFIIA initiating transcription. Our results suggest that, in TFIID, several distinct TBP/TAF interactions exist, which are formed to forestall unwanted TFIID/DNA interactions which could otherwise lead for instance to cryptic transcription initiation on genomic regions that do not contain promoter elements. In addition to the TAF1-

**Table 5.** Interaction surfaces in TBP complexes

| Interactor | Interface (Å²) |
|---|---|
| TBP dimer | 3010.2 |
| DNA | 4020.1 |
| TAF1-TAND (*D. melanogaster*) | 3287.8 |
| TAF1-TAND (Yeast) | 7483.1* |
| Mot1 (*E. cuniculi*) | 4300.0 |
| TAF11/TAF13 | 3305.2 |

Calculated with PyMol v1.8.2.0 (www.pymol.org).
*Includes TAND1 and TAND2.
DOI: https://doi.org/10.7554/eLife.30395.021

dependent inhibited canonical state, we propose an alternative inhibitory TFIID state in which the TAF11/TAF13 HF pair blocks TBP from binding TATA-box containing promoter DNA (*Figure 7*). This alternative inhibited state may serve as a further point of transcriptional control, possibly depending on promoter context or additional gene regulatory factors bound. Our results imply that, in a given TFIID complex, the TAF1-dependent and the TAF11/TAF13 HF pair-dependent TBP-blocking activities are mutually exclusive, but they may compete with each other to ascertain full blocking activity. Interestingly, however, it appears that this TAF11/TAF13 HF pair-dependent TBP binding/blocking activity is essential/required for normal TFIID function, because when we interfered in the TBP binding through mutating the CTID, yeast growth was compromised at the non-permissive conditions. At the same time, TFIID integrity was not compromised by the mutations. It is not clear at the moment whether or not the TAF1-dependent and the TAF11/TAF13 HF pair-dependent TBP blocking activities are really competing with each other, or would be simply part of a step-wise TFIID conformational change, or 'activation', process that would allow TFIID to bind to DNA only when open promoter structures would become available. Further experiments will be needed to answer these exciting questions. Transcription activators and chromatin remodeling factors may direct inhibited TFIID to specific promoters, which could be poised to be transcribed by histone H3K4 trimethylation, and alleviate the TBP-blocking through TAF-interactions or by TAF-chromatin mark interactions. Alternatively, it is conceivable that once TFIID is brought to a promoter by interactions with transcription activators and positive chromatin marks (i.e. histone H3K4me3), DNA and TFIIA together may synergize to liberate the TATA-box-binding surface of TBP from the inhibitory TAF-interactions.

The general roles of individual TAFs and the holo-TFIID complex are increasingly better understood, the mechanisms by which the cell assembles this essential multiprotein complex, however, remains largely enigmatic. The existence of discrete TFIID subassemblies containing a subset of TAFs, such as nuclear core-TFIID and the TAF2/TAF8/TAF10 complex present in the cytoplasm, provides evidence that holo-TFIID may be assembled in a regulated manner in the nucleus from preformed submodules (*Bieniossek et al., 2013*; *Gupta et al., 2016*; *Trowitzsch et al., 2015*). We analyzed TAF11-containing complexes by co-IP experiments from the cytoplasm and the nucleus of HeLa cells. In cytosolic co-IPs, we identified the TAF11/TAF13 histone fold pair, suggesting that this TFIID submodule may also be preformed in the cytoplasm (*Figure 5*). Note, however, that this cytoplasmic TAF11/TAF13 building block did not contain detectable amounts of TBP in our experiments, suggesting that trimeric TAF11/TAF13/TBP, within TFIID or as a discrete TFIID submodule, would be formed in the nucleus. Interestingly, we also identified TAF7 associated with TAF11/TAF13 in the cytoplasm, hinting at putative novel interactions between TAF11/TAF13 and TAF7. TAF1 forms stable complexes with both TAF7 and TBP (*Gupta et al., 2016*). We speculate that cytoplasmic TAF7/TAF11/TAF13 may represent an assembly intermediate toward a TAF1/TAF7/TAF11/TAF13/TBP module, which may integrate into a core-TFIID and TAF2/TAF8/TAF10 containing '8TAF' assembly (*Trowitzsch et al., 2015*) in the formation pathway to the complete nuclear holo-TFIID complex. In this TAF1/TAF7/TAF11/TAF13/TBP module, TBP would be tightly bound to either TAF1 or TAF11/TAF13, which could serve to ascertain that this putative TFIID submodule is efficiently blocked from any potentially detrimental interactions with DNA until holo-TFIID formation is completed.

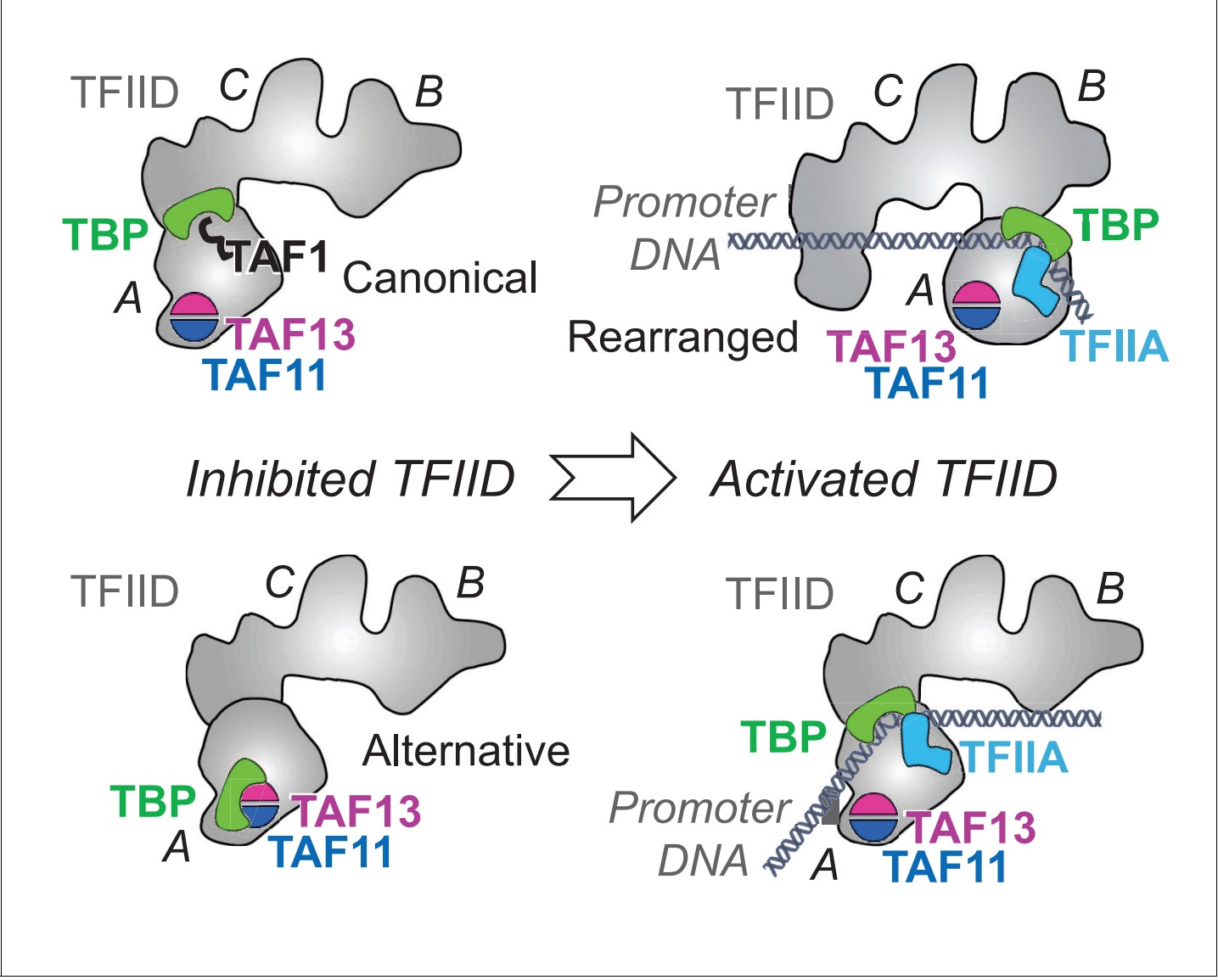

**Figure 7.** Novel TFIID regulatory state comprising TAF11/TAF13/TBP. . TFIID is thought to exist in an inhibited 'canonical' state with TAF1-TAND bound to TBP's DNA binding surface (bottom left). Activated states of TFIID (right) bind promoter DNA stabilized by TFIIA (*Cianfrocco et al., 2013*; *Louder et al., 2016*; *Papai et al., 2010*). Our results suggest a novel, alternative TFIID inhibited state comprising TAF11/TAF13/TBP (top left). TFIID is shown in a cartoon representation based on previous EM studies (*Cianfrocco et al., 2013*; *Louder et al., 2016*; *Papai et al., 2010*). TFIID lobes A, B and C are indicated. TAF11/TAF13 are placed in lobe A as suggested by immune-labeling analysis (*Leurent et al., 2002*). Promoter DNA is colored in grey.

DOI: https://doi.org/10.7554/eLife.30395.022

In the nuclei of human cells, IPs utilizing antibodies against several different TFIID specific TAFs co-precipitated all known TFIID subunits, although with variable stoichiometry. Strikingly, stoichiometry analyses carried out by NSAF calculations of our nuclear anti-TAF IPs indicated that TBP was only present in less than half of the TFIID specimens, when compared to TAF1 or TAF7 for example, suggesting that TFIID-type complexes may exist which do not contain TBP. TBP is thought to be highly mobile structurally in the context of holo-TFIID, with recent cryo-EM studies predicting large-scale migration of TBP within distinct TFIID conformational states (*Cianfrocco et al., 2013*; *Louder et al., 2016*). Our observations point to an additional level of compositional TBP and/or TAF dynamics in TFIID formation, raising the interesting possibility that the accretion of TBP in holo-TFIID

may be regulated by partially assembled nuclear TFIID building blocks including for instance the TAF11/TAF13/TBP complex we identified.

# Materials and methods

## Key resource table

| Reagent type (species) or resource | Designation | Source or reference | Identifiers | Additional information |
|---|---|---|---|---|
| gene (*Homo Sapiens* TAF11) | TAF11 | This paper | TAF11_uniprot:Q15544 | Synthesized by Genescript and cloned with his-tag as described in methods. |
| gene (*Homo Sapiens* TAF13) | Taf13 (WT; Mutant A; Mutant B) | This paper | TAF13_uniprot:Q15543 | Synthesized by Genescript and cloned with hisTag as described in methods. |
| gene (*Homo Sapiens* TBP) | TBP;TBPc | This paper | TBP_uniprot:P20226 | Synthesized by Genescript and cloned with hisTag as described in methods. |
| gene (*Homo Sapiens* TAF1) | TAF1; TAF1_TAND | This paper | TAF1_uniprot:P21675 | Synthesized by Genescript and cloned with MBP tag as described in methods. |
| gene (*Homo Sapie* TFIIAαβ) | TFIIA; TFIIAαβ | This paper | TFIIAαβ_unirprot: P52655 | Synthesized by Genescript and cloned with his tag as described in methods. |
| gene (*Homo Sapiens* TFIIAγ) | TFIIA; TFIIAγ | This paper | TFIIAγ_uniprot: P52657 | Synthesized by Genescript and cloned with his tag as described in methods. |
| gene (*Saccharomyces cerevisiae* Taf13) | Taf13 (WT; Mutant A; Mutant B) | This paper | TAF13_uniprot:P11747 | Synthesized by Genescript and cloned with hisTag as described in methods. |
| strain, strain background is BY4741 | TSA797 | EuroSCARF SRD GmbH, Germany | Y41183 | Tested for growth at permissive and non-permissive temperatures exhibiting expected phenotype. |
| strain, strain background is BY4741 | TSA636 | EuroSCARF SRD GmbH | Y41075 | Tested for growth at permissive and non-permissive temperatures exhibiting expected phenotype. |
| strain, strain background | Taf13-AID auxin-inducible degron strain | *Warfield et al., 2017* | | Prof. Steven Hahn Lab (http://research.fhcrc.org/hahn/en.html) |
| cell line (*Spodoptera frugiperda* 21) | Sf21 | Invitrogen, Carlsbad, CA, USA | | |
| cell line (Henrietta Lachs) | HeLa | Mycoplasma-free HeLa cells obtained from Betty Heller, IGBMC cell line resource www.igbmc.fr | | The cell line used was Hela WS, also called HeLa S3 or HeLa CCL-2.2 (RRID:CVCL_0058). The STR Profile report (SOH32553) stated that the "submitted sample (STRA0021) is an exact match to ATCC cell line HeLa CCL-2.2. The mycoplasma contamination of this cell line is regularly tested using the VenorTMGeM Mycoplasma Detection Kit (from Sigma Aldrich,Catalog Number MP0025). The used cells were mycoplasma free. |
| antibody | Rabbit anti-HA antibody | Sigma Aldrich | H6908 | |
| antibody | Rabbit anti-Taf5 | From Prof Tony Weil, Vanderbuilt University | | |
| antibody | Rabbit anti-Taf6 | From Prof Tony Weil, Vanderbuilt University | | Dilution 1:1000 |
| antibody | Rabbit anti-Taf8 | From Prof Tony Weil, Vanderbuilt University_ | | Dilution 1:2000 |
| antibody | Rabbit anti-Taf11 | From Prof Tony Weil, Vanderbuilt University | | Dilution 1:1000 |
| antibody | Rabbit anti-TBP | From Prof Tony Weil, Vanderbuilt University_ | | Dilution 1:2000 |
| antibody | Goat anti-rabbit HRP | Thermo Fisher Scientific Inc. | 31463 | Dilution 1:10000 |

*Continued on next page*

*Continued*

| Reagent type (species) or resource | Designation | Source or reference | Identifiers | Additional information |
|---|---|---|---|---|
| recombinant DNA reagent | MultiBac system | *Berger et al. (2004)* | | |
| recombinant DNA reagent | MutiBacTAG system | *Koehler et al. (2016)* | | |
| chemical compound, drug | DiAzKs | *Koehler et al. (2016)* | | |
| software, algorithm | XDS | *Kabsch, 2010* | | |
| software, algorithm | PHASER | *McCoy (2007)* | | |
| software, algorithm | CCP4 suite | *Winn et al. (2011)* | | |
| software, algorithm | ATSAS | *Petoukhov et al. (2012)* | | |
| software, algorithm | MSCovert | *Kessner et al. (2008)* | | |
| software, algorithm | Xi software | ERI Edinburgh | | |
| software, algorithm | HADDOCK | *de Vries et al. (2010)* | | |
| software, algorithm | MassLynx | Waters | | |
| software, algorithm | Mass Hunter | Agilent Technologies Inc. | | |
| software, algorithm | HD Examiner | Sierra Analytics Inc. | | |
| DNA for EMSA | AdMLP TATA-DNA | This paper | | Synthesized by BioSpring GmbH |

## DNA constructs

TAF11, TAF13 and TAF1-TAND were cloned in MultiBac baculovirus/insect cell transfer plasmids (*Berger et al., 2004*; *Fitzgerald et al., 2006*) and TBP$^{fl}$, TBP$^c$ and TFIIA$^{s-c}$ expression constructs were cloned in *E. coli* expression plasmids. Constructs for genetic code expansion using the Multi-BacTAG system were generated by PCR as described (*Koehler et al., 2016*). All constructs were verified by DNA sequencing.

Coding sequences of full-length TAF11 (Uniprot accession number Q15544) and TAF13 (Uniprot accession code Q15543) were synthesized at GenScript (New Jersey, USA). TAF11 contained an N-terminal hexahistidine tag spaced by a restriction site for Tobacco Etch Virus (TEV) NIA protease. TAF13 was cloned into MCS1 of the pFL acceptor plasmid from the MultiBac(*Berger et al., 2004*) suite via restriction sites NsiI and XhoI. TAF11 was inserted into MCS2 of the pFL-TAF13 plasmid via restriction sites RsrII and EcoRI.

The coding sequences for TFIIA α(AA2-59; Uniprot accession number P52655), β(AA302-376; Uniprot accession number P52655) and γ(AA2-110; Uniprot accession number P52657) were arranged into a single open reading frame by adding DNA sequences encoding for linkers L1(-DGKNTAN-SANTNTV-) and L2(-SRAVDGELFDT-). A C-terminal hexahistidine-tag was added to facilitate purification. The complete coding sequence was inserted into the bacterial expression plasmid pET28a via restriction sites NcoI and XhoI.

TBP full-length (UniProt accession number P20226) was synthesized (Genscript, New Jersey) and cloned via restriction enzymes NdeI and KpnI into a pET28a plasmid containing a hexahistidine-tag with a TEV cleavage site. TBP core (AA 155–335, UniProt accession number P20226) was generated from this plasmid by polymerase chain reaction (PCR).

The TAF1-TAND coding sequence (AA 26–168; UniProt accession number P21675) was cloned into a modified pUCDM vector coding for an engineered N-terminal TEV-cleavable maltose-binding protein (MBP) tag using sequence and ligation-independent cloning (SLIC) (*Li and Elledge, 2007*).

## Preparation of TAF11/TAF13 complex

The human TAF11/TAF13 complex was co-expressed in Sf21 insect cells using the MultiBac system (*Berger et al., 2004*). DNA encoding for an N-terminal hexa-histidine tag and a protease cleavage site for tobacco etch virus (TEV) NIa protease was added to the 5' end of the TAF11 open-reading frame and cloned into pFL plasmid (*Berger et al., 2004*). Cells were resuspended in Talon Buffer A

(25 mM Tris pH 8.0, 150 mM NaCl, 5 mM imidazole with complete protease inhibitor (Roche Molecular Biochemicals). Cells were lysed by freeze-thawing (three times), followed by centrifugation at 40,000 g in Ti70 rotor for 60 min to clear the lysate. TAF11/TAF13 complex was first bound to talon resin, pre-equilibrated with Talon Buffer A, followed by washes with Talon Buffer A, then Talon Buffer HS (25 mM Tris pH 8.0, 1M NaCl, 5 mM imidazole and complete protease inhibitor) and then again Talon Buffer A. TAF11/TAF13 complex was eluted using Talon Buffer B (25 mM Tris pH 8.0, 150 mM NaCl, 200 mM imidazole and complete protease inhibitor). Fractions containing the TAF11/TAF13 complex were dialyzed overnight against HiTrapQ Buffer A (50 mM Tris pH 8.0, 150 mM NaCl, 5 mM β-ME and complete protease inhibitor). Complex was further purified using ion exchange chromatography (IEX) with a HiTrapQ column pre-equilibrated with HiTrapQ Buffer A. After binding, column was washed with HiTrapQ Buffer A and TAF11/TAF13 eluted using a continuous gradient of HiTrapQ Buffer B (50 mM Tris pH 8.0, 1M NaCl, 5 mM β-ME and complete protease inhibitor) from 0% to 50%, followed by a step gradient to 100%. The complex was further purified by size exclusion chromatography (SEC) with a SuperdexS75 10/300 column in SEC buffer (25 mM Tris pH 7.5, 300 mM NaCl, 1 mM EDTA, 1 mM DTT and complete protease inhibitor). Mutants of TAF13 were generated by self-SLIC reaction (*Haffke et al., 2013*) and complexes expressed and purified as wild-type.

## Preparation of TBP

Full-length human TBP with an N-terminal oligo-histidine tag was expressed in *E. coli* BL21 (DE3) STAR cells at 30°C. Cells were lysed in Talon Buffer A by using a French press. Lysate was cleared by centrifugation at 40,000 g for 60 min. TBP^fl was eluted from TALON resin with Talon Buffer B using a continuous gradient. The tag was removed by TEV protease cleavage during dialysis overnight into Dialysis Buffer (25 mM Tris pH 8.0, 300 mM NaCl, 5 mM β-ME) and a reverse IMAC step was used to remove uncleaved protein. TBP^fl was polished using a SuperdexS75 16/60 equilibrated in SEC Buffer. The conserved TBP core (TBP^c) was expressed in *E. coli* Rosetta (DE3) cells at 18°C and purified as described for TBP^fl.

## Preparation of double-stranded TATA-box containing promoter DNA substrate

AdMLP) TATA-DNA was prepared from synthetic olgonucleotides d(ctgctataaaaggctg) and d(cagccttttatagcag) (BioSpring GmbH) by mixing the complementary strands in equimolar amounts in Annealing Buffer (10 mM Tris pH 8.0, 50 mM KCl, 5 mM MgCl$_2$), heating to 96°C for 2 min and slow-cooling to room temperature.

## Design and production of TFIIA$^{s-c}$

A construct encoding human TFIIA was prepared by structure-based design starting from the TFIIA/TBP/DNA structure (PDB ID 1NVP) by introducing a linker (L1) with sequence DGKNTANSANTNTVP between the TFIIA γ chain and the α chain. Similarly, a second linker (L2) with sequence SRAVDG-ELFDT was introduced connecting the α chain with the β chain giving rise to a single-chain TFIIA$^{s-c}$ construct γ-L1-α-L2-βencompassing 240 amino acid residues in total. The gene encoding for TFIIA$^{s-c}$ was cloned in a pET28a plasmid resulting in frame with a C-terminal hexa-histidine tag. TFIIA$^{s-c}$ was expressed in *E. coli* BL21 (DE3) cells at 18°C. Cells were lysed using a French press in Binding Buffer (20 mM TrispH 7.4, 150 mM NaCl and complete protease inhibitor). The lysate was cleared by centrifugation at 40,000 g for 45 min, and loaded on a Talon affinity column. After 10 column volumes of washing with Binding Buffer, TFIIA$^{s-c}$ was eluted using Elution Buffer (20 mM Tris pH 7.4, 150 mM NaCl and 250 mM imidazole). Eluted protein was dialyzed overnight in 20 mM Tris pH 7.4, 150 mM NaCl, 0.5 mM EDTA and 1 mM DTT and loaded onto a Heparin column. TFIIA$^{s-c}$ was eluted with HS Buffer (20 mM Tris pH 7.4, 1M NaCl, 0.5 mM EDTA and 1 mM DTT) applying a gradient, and polished by SEC with a Superdex75 column equilibrated in SEC Buffer 2 (20mM Tris pH 7.4, 150 mM NaCl, 0.5 mM EDTA and 1 mM DTT). Purified TFIIA$^{s-c}$ protein was aliquoted and stored in Storage Buffer (20 mM Tris pH 8.0, 1 mM DTT, 0.5 mM EDTA and 150 mM NaCl) at −80°C.

## X-ray crystallography

Large crystals of TFIIA[s-c] were obtained by vapor diffusion at room temperature from a protein solution concentrated to 15 mg/ml in Storage Buffer and equilibrated against 20 mM Tris pH 8.0 with 25 mM NaCl in the reservoir. Best crystals were obtained by streak-seeding with the TFIIA[s-c] crystals spontaneously formed in the Eppendorf tube used for storing the protein. Crystals were harvested and mounted using perfluoropolyether (PFO-X175/08) as cryo-protectant. X-ray diffraction data were collected using a Pilatus 6M detector at beamline ID29 at the European Synchrotron Radiation Facility (ESRF). TFIIA[s-c] crystals diffracted to 2.4 Å resolution. Data were processed using XDS (*Kabsch, 2010*). The structure was determined by molecular replacement (MR) utilizing software PHASER (*McCoy, 2007*), with the TFIIA crystal coordinates from the human TBP/TFIIA/TATA-DNA structure (*Bleichenbacher et al., 2003*) used as a search model. Iterative cycles of refinement and model building were performed using REFMAC5 and COOT from the CCP4 suite (*Winn et al., 2011*). Residues 2–210 could be modeled unambiguously in the electron density maps. Refinement converged at R and $R_{free}$ values of 0.18 and 0.24, respectively. Refinement statistics are provided in *Table 1*. TFIIA[s-c] coordinates and structure factors were submitted to the PDB (5M4S).

## Band shift assay

Samples for electrophoretic mobility shift assay were prepared by mixing annealed dsDNA (2 μM) with TBP[c] (4 μM) or TBP[fl] (4 μM), respectively, and TFIIA[s-c] (6 μM) in EMSA Reaction Buffer (10 mM Tris pH 8.0, 60 mM KCl, 10 mM $MgCl_2$, 10% glycerol, 2.5 mM DTT). Purified TAF11/TAF13 was added to aliquots of this mix at increasing concentrations (2μM to 64μM) followed by 1.5 hr incubation on ice. Samples were analyzed by non-denaturing 5% polyacryl-amide gel electrophoresis (PAGE) using EMSA Running Buffer (25 mM Tris, 190 mM Glycine, 5 mM Mg Acetate, pH 8.0). Gels were stained with ethidium bromide (EtBr).

## Preparation of TAF11/TAF13/TBP complex

TAF11/TAF13 (wild-type and mutants) was mixed with TBP in a 1:1.1 molar ratio (40 μM total protein) in Complex Reaction Buffer (25 mM Tris pH 8.0, 300 mM NaCl, 1 mM EDTA, 1 mM DTT, Leupeptin, Pepstatin and complete protease inhibitor) and incubated on ice for 90 min, followed by SEC with a SuperdexS200 10/300 column pre-equilibrated in Reaction Buffer.

## Analytical ultracentrifugation

Purified TAF11/TAF13/TBP complex was analyzed by analytical ultracentrifugation (AUC) in an An-60Ti rotor in a Beckman XL-I analytical ultracentrifuge (Beckman Coulter, Brea, CA) at 42,000 rpm at 10°C for 16 hr. Data were analyzed with software Sedfit (*Schuck, 2000*).

## Native mass spectrometry

Proteins and complexes were buffer exchanged into 150 mM ammonium acetate pH7.5 before MS analysis using Vivaspin 10 kDa MWCO (Sartorius, Goettingen, Germany). 2 ul of the protein solution was then nano-electrosprayed from a gold-coated borosilicate glass capillaries made in the Robinson lab (*Hernández and Robinson, 2007*). All mass spectrometry measurements were performed on a QToF instrument optimized for high mass measurements in positive ion mode. MS spectra were recorded at capillary and cone voltages of 1.7 kV and 80 V, respectively. Other instrument parameters were ToF pressure $1.76 \times 10^6$ mbar and analyser pressure of $2 \times 10^4$ mbar. For the collision-induced dissociation the collision energy was increased up to 100 V to induce dissociation. All mass spectra were calibrated using an aqueous solution of caesium iodide and data were processed using MassLynx software V4.1.

## TAF1-TAND production and competition assay

Human TAF1-TAND (*Liu et al., 1998*; *Mal et al., 2004*) containing an N-terminal maltose-binding protein (MBP) tag was expressed in Sf21 insect cells using MultiBac (*Berger et al., 2004*). Cells were lysed in MBP Binding Buffer (20 mM Tris pH 8.0, 300 mM NaCl, 1 mM EDTA, 1 mM DTT, complete protease inhibitor) by freeze-thawing, followed by centrifugation at 20,000 rpm in a JA25.5 rotor for 45 min. MBPTAF1-TAND was bound to amylose resin pre-equilibrated in MBP Binding Buffer, followed by extensive washing (20 column volumes). MBPTAF1-TAND loaded resin was incubated with

an excess of preformed TAF11/TAF13/TBP complex for 60 min at 4℃. The column was washed and flow-through as well as wash fractions were collected. Bound protein(s) were eluted using MBP Elution buffer (20 mM Tris pH 8.0, 300 mM NaCl, 1 mM EDTA, 1 mM DTT, 10 mM Maltose, complete protease inhibitor). Samples were analyzed by SDS-PAGE followed by staining with Coomassie Brilliant Blue (Sigma Aldrich).

## Small-angle X-ray scattering experiments

Small-angle X-ray scattering (SAXS) experiments were carried out at the ESRF BioSAXS beamline BM29 (*Pernot et al., 2013*). 30 µl of each of purified TAF11/TAF13/TBP, TAF11/TAF13 and TBP at a range of concentrations (*Table 3*) and SAXS Sample Buffer (25 mM Tris pH 8.0, 300 mM NaCl, 1 mM EDTA, 1 mM DTT and complete protease inhibitor) were exposed to X-rays and scattering data collected using the robotic sample handling available at the beamline. Ten individual frames were collected for every exposure, each 2 s in duration, using the Pilatus 1M detector (Dectris AG). Data were processed with the ATSAS software package (*Petoukhov et al., 2012*). Individual frames were processed automatically and independently within the EDNA framework (*Incardona et al., 2009*), yielding individual radially averaged curves of normalized intensity versus scattering angle $S = 4\pi Sin\theta/\lambda$. Additional data reduction within EDNA utilized the automatic data processing tools of ATSAS software package, to combine timeframes, excluding any data points affected by aggregation induced by radiation damage, yielding the average scattering curve for each exposure series. Matched buffer measurements taken before and after every sample were averaged and used for background subtraction. Merging of separate concentrations and further analysis steps were performed manually using PRIMUS (*Petoukhov et al., 2012*). Forward scattering (I(0)) and radius of gyration (Rg) were calculated from the Guinier approximation, to compute the hydrated particle volume using the Porod invariant and to determine the maximum particle size (Dmax) from the pair distribution function computed by GNOM (*Petoukhov et al., 2012*). 40 *ab initio* models were calculated for each sample, using DAMMIF (*Petoukhov et al., 2012*), and then aligned, compared and averaged (evidencing minimal variation) using DAMAVER (*Petoukhov et al., 2012*). The most representative model for TAF11/TAF13/TBP and TAF11/TAF13 selected by DAMAVER were compared to each other as well as the known structure of TBP, with overlays of the resulting models generated in PyMOL. The fits to the experimental data of the models and the theoretical scattering of the calculated structures were generated with CRYSOL (*Petoukhov et al., 2012*).

## Cross-linking/mass spectrometry (CLMS) experiments

### BS3 cross-linking

Purified TAF11/TAF13/TBP and TAF11/TAF13 complexes were cross-linked separately by BS3 at complex/BS3 ratio of 1:25 [w/w] in Cross-linking Buffer (25 mM HEPES, pH 8.0, 300 mM NaCl, 1 mM DTT, 1 mM EDTA and complete protease inhibitor) for 2 hr on ice. The reaction was quenched by adding saturated ammonium bicarbonate solution followed by incubation on ice (45 min). Cross-linked samples were then further purified by injecting on a SuperdexS200 10/300 column. Peak fractions containing purified cross-linked samples were concentrated using pin concentrators (Amicon) and separated by SDS-PAGE using a NuPAGE 4–12% bis-Tris gel (Thermo Fisher Scientific).

The gel bands corresponding to cross-linked complexes were reduced, alkylated and trypsin digested following standard procedures (*Maiolica et al., 2007*) and digested peptides were desalted using C18 StageTips (*Rappsilber et al., 2007*). Peptides were analyzed on an LTQ Orbitrap Velos mass spectrometer (Thermo Fisher Scientific) that was coupled with a Dionex Ultimate 3000 RSLC nano HPLC system. The column with a spray emitter (75 µm inner diameter, 8 µm opening, 250 mm length; New Objectives) was packed with C18 material (ReproSil-Pur C18-AQ 3 µm; Dr Maisch GmbH, Ammerbuch-Entringen, Germany) using an air pressure pump (Proxeon Biosystems) (*Ishihama et al., 2002*). Mobile phase A consisted of water and 0.1% formic acid. Mobile phase B consisted of 80% acetonitrile and 0.1% formic acid. Peptides were loaded onto the column with 2% B at 500 nl/min flow rate and eluted at 200 nl/min flow rate with two gradients: linear increase from 2% B to 40% B in 90 min; then increase from 40% to 95% B in 11 min. The eluted peptides were directly sprayed into the mass spectrometer.

Peptides were analyzed using a high/high strategy (*Chen et al., 2010*): both MS spectra and MS2 spectra were acquired in the Orbitrap. FTMS full scan spectra were recorded at 100,000 resolution.

The eight highest intensity peaks with a charge state of three or higher were selected in each cycle for fragmentation. The fragments were produced using CID with 35% normalized collision energy and detected by the Orbitrap at 7500 resolution. Dynamic exclusion was set to 90 s and repeat count was 1. Peak lists were generated by MSCovert (ProteoWizard version 3.0.6618) (*Kessner et al., 2008*).

## DiAzKs cross-linking

An unnatural amino acid, DiAzKs, was introduced at K34 position of TAF13 using genetic code expansion (GCE) we implemented recently in our baculovirus/insect cell system (MultiBacTAG) (*Koehler et al., 2016*). TAF11/TAF13-K34DiAzKs was purified similarly as wild type. Briefly, TAF11/TAF13-K34DiAzKs (as well as wild type) and TBP were mixed in 1:1.25 molar ratio in Incubation Buffer (25 mM Tris, pH 8.0, 300 mM NaCl, 1 mM DTT, 1 mM EDTA and complete protease inhibitor) and incubated on ice for 2 hr. Reactions were then split into two aliquots. One aliquot was UV irradiated for 15 min on ice using a 345 nm filter with an approximately 40 cm distance to the 1000 W lamp. Cross-linked samples were then separated on SDS-PAGE using a NuPAGE 4–12% bis-Tris gel (Thermo Fisher Scientific). Gel bands were processed as above for BS3 CLMS.

Peptides were analyzed on an Orbitrap Fusion Lumos Tribrid mass Spectrometer (Thermo Fisher Scientific) coupled to a Dionex UltiMate 3000 RSLC nano HPLC system using a 75 µmx50cm PepMap EASY-Spray column (Thermo Fisher Scientific). Eluted peptides were directly sprayed into the mass spectrometer through EASY-Spray source (Thermo Fisher Scientific) and analyzed using a high/high strategy (*Chen et al., 2010*): both MS spectra and MS2 spectra were acquired in the Orbitrap. MS1 spectra were recorded at 120,000 resolution and peptides with charge state of 3 to 8 were selected for fragmentation at top speed setting. The fragments were produced using HCD with 30% normalized collision energy and detected by the Orbitrap at 15000 resolution. Dynamic exclusion was set to 60 s and repeat count was 1. Peak lists were generated by MaxQuant (version 1.5.3.30) (*Cox and Mann, 2008*) at default parameters except for 'top MS/MS peaks per 100 Da' being set to 100. Cross-linked peptides were matched to spectra using Xi software (ERI, Edinburgh).

## Hydrogen-deuterium exchange/mass spectrometry

Hydrogen-deuterium exchange/mass spectrometry (HDX-MS) experiments were fully automated using a PAL autosampler (CTC Analytics). This controlled the start of the exchange and quench reactions, the proteolysis temperature (4°C), the injection of the deuterated peptides, as well as management of the injection and washing valves; it also triggered the acquisition of the mass spectrometer and HPLC and UPLC pumps. A Peltier-cooled box (4°C) contained two Rheodyne automated valves, a desalting cartridge (Trap Acquity UPLC Protein BEH C18 2.1 × 5 mm,Waters) and a UPLC column (Acquity UPLC BEH C18 1.7 µm 1 × 100 mm, Waters). HDX-MS reactions were carried out using either TAF11/TAF13 or TBP alone or in complex at a concentration of 20 µM. Deuteration was initiated by a 5-fold dilution of the protein samples (10 µl) with the same buffer in $D_2O$ (40 µl). The proteins were deuterated for 15 s or 2 min at 4°C. The exchange was quenched using 50 µl of 200 mM glycine-HCl, pH 2.5 at 4°C. The proteins or complexes were digested online with immobilized porcine pepsin (Sigma) and recombinant nepenthesin-1. The peptides were desalted for 6 min using a HPLC pump (Agilent Technologies) with 0.1% formic acid in water, at a flow rate of 100 µl/min. Desalted peptides were separated using a UPLC pump (Agilent Technologies) at 50 µl/min for 10 min with 15–70% gradient B (Buffer A: 0.1% formic acid in water; Buffer B: 0.1% formic acid in 95% acetonitrile), followed by 1 min at 100% B. The peptide masses were measured using an electrospray-TOF mass spectrometer (Agilent 6210) in the 300–1300 m/z range. Each deuteration experiment was conducted in triplicate. The Mass Hunter (Agilent Technologies) software was used for data acquisition. The HD Examiner software (Sierra Analytics) was used for HDX-MS data processing. Identification of peptides generated by the digestion was done as described previously (*Giladi et al., 2016*). Different proteases (pepsin, nepenthesin-1, nepenthesin-2, rhizopuspepsin) or their combinations were tested for protein digestion with pepsin-nepenthesin-1 pair providing the best digestion parameters and sequence coverage.

## Integrative multiparameter-based model building and refinement

Initial models of the two component structures (TAF11/TAF13, TBP) were taken from the PDB (1BH8 and 1CDW) (*Birck et al., 1998*; *Nikolov et al., 1996*). 1BH8 was extended to include a helix structure missing from the complete histone-fold domain as described before (*Birck et al., 1998*). The structure of the complex was constructed in a two-stage workflow. Initially, a model of the structured core of the complex was constructed by rigid body docking using the HADDOCK webserver (*de Vries et al., 2010*). The resulting complex structures and their scores were visually analyzed against the SAXS data to select the highest scoring structure that fit within the SAXS envelopes.

The selected complex with the highest scores was then refined integrating the cross-linking data. The HADDOCK complex was used as an input to MODELLER 9.14 (*Webb and Sali, 2014*) with the complete sequences (including loop structures). Observed cross-links were included as restraints in the refinement with a mean distance of 11.4 Å. Refinement was performed iteratively until more than 90% of all distance constraints could be accommodated while maintaining the fit to the SAXS envelope.

## Cell growth experiments

Yeast Taf13 wild-type (WT), as well as Mutant A and Mutant B, were cloned along with native promoters into the LEU2 (auxotrophic marker) containing plasmid pRS415 (Genscript Corp., Piscataway, NJ) by using the BamHI and NotI restriction enzyme sites. Constructs thus generated were transformed into yeast strain BY4741 (comprising endogenous wild-type Taf13) as well as the temperature-sensitive (ts) yeast strains TSA797 (ts *taf13*) and TSA636 (ts *taf13*) (EuroSCARF, SRD GmbH, Germany). Transformed yeast containing the plasmids were restreaked onto selective media and grown at permissive (30°C) or non-permissive (37°C) temperatures, and plates imaged. To determine growth rates, ts strains transformed with empty vector or Taf13 expression plasmids were grown in liquid media at 37°C. In a separate experiment, the above constructs were transformed into a Taf13-AID auxin-inducible degron strain (*Warfield et al., 2017*), and grown at 30°C in liquid media supplemented with 500 μM indole-3-acetic acid (IAA) or an equivalent volume of DMSO (used to prepare IAA stocks). Empty degron strain was used as a negative control. Absorbance at 600 nm was measured every hour for all cell growth experiments. Three (ts strains) or two (degron strain) independent experiments were performed and data were normalized against the first time point taken. Average absorbance was plotted against time, standard errors of mean (SEM) were calculated over each data point.

For spot assays, overnight cultures of empty degron strain as well as degron strain transformed with Taf13 wild-type and mutant expression plasmids were washed and resuspended in milli-Q water to obtain identical densities. Serial 10-fold dilutions were spotted on solid media (YPD, or synthetic drop-out media -LEU) supplemented with 500 μM IAA or DMSO, and incubated at 30°C for 48–72 hr.

## TFIID immunoprecipitation experiments

### Yeast TFIID

Yeast Taf13 wild-type (WT) as well as Mutant A and Mutant B, along with native promoters and a hemagglutinin (HA) tag at the C-terminus, were cloned into the LEU2 (auxotrophic marker) containing plasmid pRS415 (Genscript Corp., Piscataway, NJ). Constructs were transformed into yeast strain BY4741 (comprising endogenous wild-type Taf13). Yeast cells were grown in suspension culture at 30°C and harvested in mid log phase by centrifugation. The cells were then lysed in ice cold co-IP buffer (20 mM HEPES pH7.9, 200 mM Potassium Acetate, 10% glycerol, 0.1% NP40, 1 mM DTT and complete protease inhibitor) using glass bead beating at 4°C. The lysate was centrifuged at 13,000 for 15 min at 4°C. Samples were pre-cleared by mixing with protein A-Sepharose beads (Generone) for 60 min. Pre-cleared samples were then mixed with 2 μl of rabbit anti-HA antibody (1 mg/ml; Sigma) for 2 hr at 4°C. Next, 30 μl of protein A- Sepharose beads were used to capture protein complexes from each sample by mixing for 60 min at 4°C. Captured protein complexes on beads were washed 4 times with 1 ml ice cold co-IP buffer. Then, protein gel loading buffer was added to the beads followed by heating at 98°C for 5 min. Proteins were separated by SDS-PAGE (NuPAGE 4–12% bis-Tris gel, Thermo Fisher Scientific) and then transferred to PVDF membranes. Membranes were then used for Western Blot (WB) with polyclonal anti-Taf, anti-TBP and anti-HA antibodies.

Goat anti-rabbit antibody conjugated to HRP substrate (Thermo Fisher Scientific) was used as secondary antibody. The membranes were developed with ECL WB detection reagent (Thermo Fisher Scientific).

### Human TFIID

HeLa cells were grown in suspension culture. $10^{11}$ cells were harvested by centrifugation and a nuclear extract was prepared according to a modified protocol (*Dignam et al., 1983*). Briefly, nuclei were prepared by resuspending the pellets in four packed cell volume (PCV) of 50 mM Tris, pH 7.9; 1 mM EDTA; 1 mM DTT and proteinase inhibitors and opening the cells with a Dounce-homogenizer. Nuclei were collected by centrifugation and lysed in 4 PCV of 50 mM Tris, pH 7.9; 25% glycerol; 500 mM NaCl; 0.5 mM EDTA; 1 mM DTT and protease inhibitors by powerful strokes. The lysate was centrifuged at 50,000 g for 20 min. The supernatant was filtered and proteins precipitating in 30% (w/v) $(NH_4)_2SO_4$ were pelleted. They were resuspended in 50 mM Tris, pH 7.9; 20% glycerol; 100 mM KCl; 5 mM $MgCl_2$; 1 mM DTT and dialyzed against the same buffer.

For immunopreciptation. 200 µl protein G-Sepharose (Pharmacia) was incubated with approximately 50 µg of the different antibodies (as indicated). Washed antibody-bound beads were then mixed with 4 mg of pre-cleared HeLa cell nuclear extract and incubated overnight at 4°C. Antibody-protein G-Sepharose-bound protein complexes were washed three times with IP buffer (25 mM Tris pH 7.9, 10% (v/v) glycerol, 0.1% NP40, 0.5 mM DTT, 5 mM $MgCl_2$) containing 0.5 M KCl and twice with IP buffer containing 100 mM KCl. Immunoprecipitated proteins were eluted from the protein G columns with 0.1 M glycine (pH 2.5) and quickly neutralized with 2 M Tris (pH 8.8).

For analysis by mass spectrometry, samples were reduced, alkylated and digested with LysC and trypsin at 37°C overnight. They were then analyzed using an Ultimate 3000 nano-RSLC (Thermo Fischer Scientific) coupled in line with an Orbitrap ELITE (Thermo Fisher Scientific). Briefly, peptides were separated on a C18 nano-column with a linear gradient of acetonitrile and analyzed in a Top 20 CID (Collision Induced Dissociation) data-dependent mass spectrometry. Data were processed by database searching using SequestHT (Thermo Fisher Scientific) with Proteome Discoverer 1.4 software (Thermo Fisher Scientific) against the Human Swissprot database (Release 2013_04, 20225 entries). Precursor and fragment mass tolerance were set at 7 ppm and 0.5 Da, respectively. Trypsin was set as enzyme, and up to two missed cleavages were allowed. Oxidation (M) was set as variable modification, and Carbamidomethylation (C) as fixed modification. Peptides were filtered with a 5% FDR (false discovery rate) and rank 1. Proteins were identified with one peptide.

## Accession codes

Atomic coordinates and structure factors have been deposited in the Protein Data Bank (*Source code 1*, PDB ID 5M4S). Proteomics data have been submitted to PRIDE (Accession number PXD005676)

## Acknowledgements

We thank all members of our laboratories for helpful discussions. We are grateful to Max Nanao, Aurelien Deniaud, Christian Becke, Moreno Wichert and Timothy J Richmond for valuable contributions. Frederic Garzoni is acknowledged for MultiBac expressions. We thank Steven Hahn for providing the Taf13-AID strain, Tony Weil for providing yeast Taf antibody reagents and purified yeast TFIID complex and Irwin Davidson for providing human anti-TAF11 antibodies for IPs. We thank M Joint and L Negroni for proteomic analyses. This work was supported by the Agence Nationale de Recherche (ANR, France) DiscoverIID (to LT and IB). CK and EAL are funded by the Baden-Wuerttemberg Stiftung (Germany). This work was supported by the Wellcome Trust (Senior Research Fellowship to JR: 103139, Centre core grant: 092076, instrument grant: 108504). LT is recipient of a European Research Council (ERC) Advanced grant (Birtoaction). IB is recipient of a Senior Investigator Award from the Wellcome Trust. This research received support from BrisSynBio, a BBSRC/EPSRC Research Centre for synthetic biology at the University of Bristol (BB/L01386X/1).

# Additional information

## Funding

| Funder | Grant reference number | Author |
|---|---|---|
| Baden-Württemberg Stiftung | | Edward A Lemke |
| Wellcome Trust | Senior Research Fellowship 10313 | Juri Rappsilber |
| H2020 European Research Council | ERC-2013-340551 | Làszlò Tora |
| Agence Nationale de la Recherche | ANR-13-BSV8-0021-03 | Làszlò Tora<br>Imre Berger |
| Wellcome Trust | | Imre Berger |
| Research Councils UK | | Imre Berger |

The funders had no role in study design, data collection and interpretation, or the decision to submit the work for publication.

## Author contributions

Kapil Gupta, Data curation, Formal analysis, Investigation, Methodology, Writing—original draft, Writing—review and editing, Carried out experiments, Interpreted experiments, Analyzed data, Co-wrote the manuscript with input from all authors; Aleksandra A Watson, Data curation, Formal analysis, Writing—original draft; Tiago Baptista, Data curation, Writing—original draft, Analyzed co-immuneprecipitations, Carried out cell growth experiments and Western Blots; Elisabeth Scheer, Christine Koehler, Juan Zou, Eaazhisai Kandiah, Data curation, Writing—original draft; Anna L Chambers, Data curation, Methodology, Writing—original draft; Ima Obong-Ebong, Arturo Temblador, Data curation; Adam Round, Data curation, Formal analysis, Carried out and interpreted SAXS experiments; Eric Forest, Data curation, Writing—original draft, Carried out and interpreted HDX experiments; Petr Man, Data curation, Carried out and interpreted HDX experiments; Christoph Bieniossek, Data curation, Supervision, Carried out and interpreted original TAF11/TAF13/TBP interaction studies; Ernest D Laue, Edward A Lemke, Juri Rappsilber, Carol V Robinson, Supervision; Didier Devys, Supervision, Investigation, Writing—original draft; Làszlò Tora, Conceptualization, Supervision, Funding acquisition, Validation, Investigation, Methodology, Writing—original draft, Writing—review and editing; Imre Berger, Conceptualization, Formal analysis, Supervision, Funding acquisition, Validation, Methodology, Writing—original draft, Project administration, Writing—review and editing

## Author ORCIDs

Anna L Chambers http://orcid.org/0000-0001-8133-6240
Arturo Temblador http://orcid.org/0000-0003-3076-6317
Ernest D Laue http://orcid.org/0000-0002-7476-4148
Didier Devys http://orcid.org/0000-0001-9655-3512
Làszlò Tora https://orcid.org/0000-0001-7398-2250
Imre Berger http://orcid.org/0000-0001-7518-9045

## Decision letter and Author response

Decision letter https://doi.org/10.7554/eLife.30395.027
Author response https://doi.org/10.7554/eLife.30395.028

# Additional files

## Supplementary files

• Source code 1. Crystal structure coordinates of human TFIIA$^{s-c}$
DOI: https://doi.org/10.7554/eLife.30395.023
• Transparent reporting form

DOI: https://doi.org/10.7554/eLife.30395.024

### Major datasets

The following dataset was generated:

| Author(s) | Year | Dataset title | Dataset URL | Database, license, and accessibility information |
|---|---|---|---|---|
| Gupta K, Watson AA, Baptista T, Scheer E, Chambers AL, Koehler C, Zou J, Obong-Ebong I, Kandiah E, Temblador A, Round A, Forest E, Man P, Bieniossek C, Laue ED, Lemke EA, Rappsilber J, Robinson CV, Devys D, Tora L, Berger I | 2017 | Architecture of TAF11/TAF13/TBP complex suggests novel regulatory state of human general transcription factor TFIID | https://www.ebi.ac.uk/pride/archive/projects/PXD005676 | Publicly available at EBI PRIDE Archive (accession no: PXD005676) |

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
