## [Decision Letter]

Thank you for submitting your article "Architecture of TAF11/TAF13/TBP complex suggests novel regulatory state in General Transcription Factor TFIID function" for consideration by *eLife*. Your article has been favorably evaluated by Jessica Tyler (Senior Editor) and three reviewers, one of whom is a member of our Board of Reviewing Editors. The reviewers have opted to remain anonymous.

The reviewers have discussed the reviews with one another and the Reviewing Editor has drafted this decision to help you prepare a revised submission.

Summary:

The authors report a novel role for human TAF11/TAF13 in binding to the DNA-binding face of TBP and thereby inhibiting the ability of TBP to bind DNA. The mapping of the interaction with TBP is based on competition with the TAF1 TAND domain, which binds to the concave surface of TBP, as well as on H-D exchange and mass spec/cross-linking experiments. The authors also find that TAF11/TAF13 competes with TFIIA for binding to DNA, in contrast with a previous study of the yeast orthologues showing that TAF11/TAF13 stabilizes TFIIA/TBP binding. Of two mutations identified in TAF13 that disrupt ternary complex formation in vitro, one does not support growth in a yeast assay. The authors discuss the implications of their results for regulation of TFIID assembly.

Essential revisions:

1) A significant concern is that the in vivo effect of the TAF13 mutation could be explained by the known role of TAF13 as part of TFIID, rather than by an independent role of TAF11/TAF13 in regulating TBP. Additional experiments are thus needed to support the in vivo relevance of the in vitro results. A possible approach could include affinity purification of TAF13 mutant A to examine its incorporation into TFIID, or some other experiment that would support a role for TAF11/TAF13 interactions with TBP in regulating transcription.

2) The discrepancy regarding the effects of the two TAF13 mutants is not adequately addressed and raises questions about the validity of the authors model. While the authors describe mutant B as having a less severe effect on TBP binding than mutant A, almost no difference is evident in Figure 4. Since both mutations have an equivalent effect on ternary complex formation as assayed by size-exclusion chromatography (SEC), the fact that only mutant A has a phenotype in vivo suggests that the ability of TAF13 to bind to TBP does not account for the deleterious effect of this mutation. This issue needs to be addressed, either by further characterizing the effects of the two mutations on binding affinity, testing of additional mutants, or some other approach that can more convincingly connect in vivo and in vitro behaviors.

3) The SAXS data do not add any meaningful information to the manuscript and should be removed. The SEC data already convincingly show complex formation, so the fact that the ab initio bead model has a larger volume in the presence of TBP is expected. Given the poor fit of the TAF11/TAF13/TBP model to the envelope and the large regions of uninterpretable volume, this approach unfortunately did not yield useful information.

---

## [Author Response]

Essential revisions:1) A significant concern is that the in vivo effect of the TAF13 mutation could be explained by the known role of TAF13 as part of TFIID, rather than by an independent role of TAF11/TAF13 in regulating TBP. Additional experiments are thus needed to support the in vivo relevance of the in vitro results. A possible approach could include affinity purification of TAF13 mutant A to examine its incorporation into TFIID, or some other experiment that would support a role for TAF11/TAF13 interactions with TBP in regulating transcription.

We thank the reviewers for this insightful comment that helped. We have addressed this question in the revised version of our manuscript by carrying out immuno-precipitations with anti-HA antibody to IP the HA-tagged Taf13 wild-type (wt) and Taf13 mutants. We probed HA-Taf13 (wild-type), HA-Taf13-MutA and HA-Taf13-MutB IPs with specific antibodies raised against a number of TFIID subunits and we show now in new Figure 4 that all tested Tafs and TBP efficiently co-IP with all either the wt or the mutant versions of Taf13. On these blots we tested the presence of Taf5, Taf6, Taf8, Taf11 and Taf13 in addition to TBP. As a positive control we used purified holo-TFIID. Our results demonstrate that the mutations we have introduced in Taf13 do not compromise TFIID integrity. Rather, both mutants (A and B) are incorporated in TFIID in the same way as wild-type. These new results, shown in novel Figure 4 in the revised version of our manuscript, compellingly support our model shown in Figure 7, which proposes two TBP binding modes by TAF1-NTD and by TAF11/TAF13 within TFIID, both engaging with the DNA-binding surface of TBP.

2) The discrepancy regarding the effects of the two TAF13 mutants is not adequately addressed and raises questions about the validity of the authors model. While the authors describe mutant B as having a less severe effect on TBP binding than mutant A, almost no difference is evident in Figure 4. Since both mutations have an equivalent effect on ternary complex formation as assayed by size-exclusion chromatography (SEC), the fact that only mutant A has a phenotype in vivo suggests that the ability of TAF13 to bind to TBP does not account for the deleterious effect of this mutation. This issue needs to be addressed, either by further characterizing the effects of the two mutations on binding affinity, testing of additional mutants, or some other approach that can more convincingly connect in vivo and in vitro behaviors.

We agree with the reviewers that the previous data has probably not been sufficiently clear. Therefore, have repeated the experiments shown in our previous Figure 4 in the original manuscript with higher concentration of total protein (40μm injected as compared to previously 20μm), and replaced the data shown in our previous Figure 4 with new SEC profiles and SDS-gel sections in the revised Figure 4. We reasoned that increasing sample concentration will shift the equilibrium towards complex formation for Mutant B. At the same time, given that Mutant A is virtually incompetent in TBP binding, we anticipated that increasing the concentration will have no significant effect on SEC profile and SDS-PAGE with this mutant. The new data is shown in novel Figure 4, indeed evidencing a clear difference between Mutant A (not binding) and Mutant B (residual binding, but different from wt) in SEC profile and SDS-PAGE. Highlighting this observation, we boxed Coomassie-stained TBP co-migrating with TAF11/TAF13 in red in the right panel of our revised Figure 4. We conclude that Mutant B retains residual binding to TBP, which allows rescue of cell growth in both the ts and the degron strains. MutA, in contrast, results in cell growth arrest in both strains. Thus our in vitro and in vivo results are consistent.

3) The SAXS data do not add any meaningful information to the manuscript and should be removed. The SEC data already convincingly show complex formation, so the fact that the ab initio bead model has a larger volume in the presence of TBP is expected. Given the poor fit of the TAF11/TAF13/TBP model to the envelope and the large regions of uninterpretable volume, this approach unfortunately did not yield useful information.

As suggested, we have removed the SAXS data from Figure 2. However, since we have used the experimental SAXS curves for the multi-parameter calculations, we have kept the corresponding experimental curves in the supplement (novel Figure 3—figure supplement 1).